# Modulation of brain signal variability in visual cortex reflects aging, GABA, and behavior

**Poortata Lalwani[1]\*, Thad Polk[1], Douglas D Garrett[2,3]**

[1]Department of Psychology, University of Michigan, Ann Arbor, United States; [2]Max Planck UCL Centre for Computational Psychiatry and Ageing Research, Berlin, Germany; [3]Center for Lifespan Psychology, Max Planck Institute for Human Development, Berlin, Germany

**Abstract** Moment-to-moment neural variability has been shown to scale positively with the complexity of stimulus input. However, the mechanisms underlying the ability to align variability to input complexity are unknown. Using a combination of behavioral methods, computational modeling, fMRI, MR spectroscopy, and pharmacological intervention, we investigated the role of aging and GABA in neural variability during visual processing. We replicated previous findings that participants expressed higher variability when viewing more complex visual stimuli. Additionally, we found that such variability modulation was associated with higher baseline visual GABA levels and was reduced in older adults. When pharmacologically increasing GABA activity, we found that participants with lower baseline GABA levels showed a drug-related increase in variability modulation while participants with higher baseline GABA showed no change or even a reduction, consistent with an inverted-U account. Finally, higher baseline GABA and variability modulation were jointly associated with better visual-discrimination performance. These results suggest that GABA plays an important role in how humans utilize neural variability to adapt to the complexity of the visual world.

**\*For correspondence:**
poortata@umich.edu

**Competing interest:** The authors declare that no competing interests exist.

## Editor's evaluation

This study by Lalwani et al. combines multiple complementary neuroscientific methods to understand the neural response to visual stimulus complexity in the human brain across lifespan. Lalwani and colleagues provide important and convincing evidence, drawing from appropriate and validated methodology. The manuscript has been comprehensively revised and now includes key control analyses, improved content throughout and clarifications. The study is of broad interest to neuroscientists and biologists interested in aging and sensory processing.

## Introduction

Our sensory modalities are constantly processing myriad inputs that differ in complexity across various dimensions. And yet, our neural networks can accommodate and process such heterogeneity. It has been postulated that an efficient neural network may do this by modulating neural dynamics to align with the complexity of input stimuli; neural dynamics would be increased to process more complex stimuli, but compressed to process less complex stimuli (*Hermundstad et al., 2014*; *Orbán et al., 2016*; *Garrett et al., 2020*). One neural measure that scales with stimulus complexity and captures this ability is moment-to-moment variability in the blood oxygen level-dependent signal ($SD_{BOLD}$). In fact, research so far has demonstrated that $SD_{BOLD}$ is a far more powerful predictor of cognitive abilities than the traditionally used mean-BOLD signal changes (*Garrett et al., 2013b*; *Garrett et al.,*

*2011*; *Grady and Garrett, 2014*) (including in the domain of stimulus selectivity *Garrett et al., 2020*). Our recent work (*Krohn et al., 2023*) shows that regional BOLD variability can even drive the strength of functional connectivity measures (another popular fMRI measure). Thus, 'local dynamics' being captured by moment-to-moment variability may be a unique and indispensable measure of human brain function in relation to how we align to the complexity of the world around us. In the current study, we examine how human aging affects our ability to align these dynamics to stimulus complexity, and the neurochemical mechanisms subserving such alignment.

Although age comparisons have not yet been made, recent research only in older adults found that the ability to upregulate visuo-cortical $SD_{BOLD}$ when viewing more complex, feature-rich visual stimuli characterized those with better visuo-cognitive performance (*Garrett et al., 2020*). These results suggest that upregulation of variability in response to the increased complexity of stimulus input (or $\overrightarrow{\Delta}SD_{BOLD}$) may be an index of the flexibility of an individual's neural system and predict individual differences in sensory processing. Moreover, previous research suggests that the modulation of variability in response to task demands is reduced in older adults compared to younger adults (*Garrett et al., 2011*; *Garrett et al., 2010*; *Garrett et al., 2013a*) and that this reduction is associated with poorer behavioral performance (*Garrett et al., 2010*; *Garrett et al., 2013a*; *Grady and Garrett, 2018*; *Waschke et al., 2021*; *Rieck et al., 2017*; *Rieck et al., 2022*). We thus hypothesized that $\overrightarrow{\Delta}SD_{BOLD}$ (expected to scale positively with visual stimulus complexity) would also be lower in older adults and that this reduction would be linked with poorer visual discrimination. But what is the basis of such age-based (and individual) differences in $\overrightarrow{\Delta}SD_{BOLD}$?

Gamma-aminobutyric acid (GABA), the brain's major inhibitory neurotransmitter, is one plausible candidate. GABA has been associated with cortical plasticity (*Jones, 1993*; *Hensch et al., 1998*; *Fagiolini et al., 2004*) with pattern complexity (*Monteforte and Wolf, 2010*) and with the dynamic range *Shew et al., 2011*; *Agrawal et al., 2018* of neural networks. Computational work has found

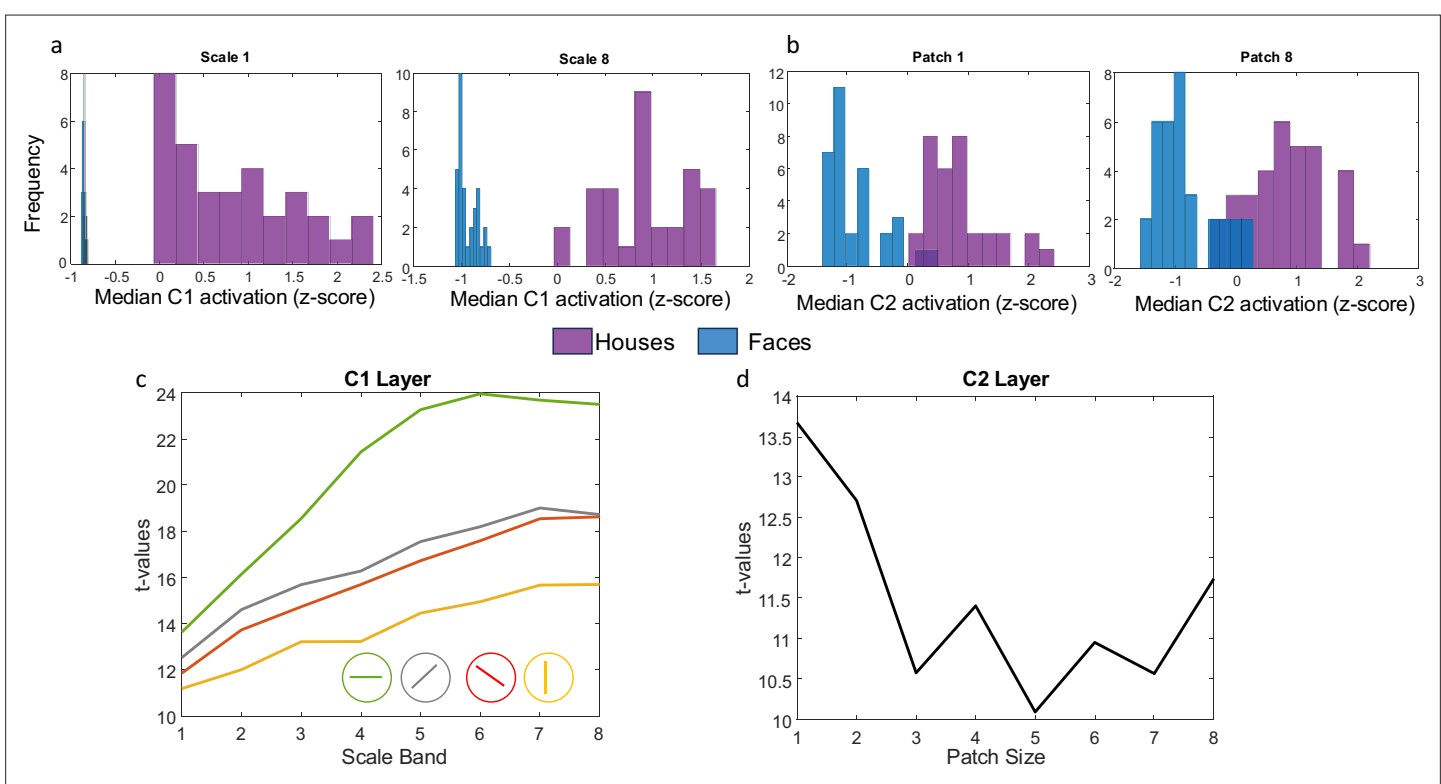

**Figure 1.** Example C1 and C2 activation distributions to house and face stimuli. (**A**) Z-scored median activation at C1 for all images in the two stimulus categories (faces in blue, houses in purple) for one orientation and two different scales. (**B**) Z-scored median activation at C2 for all images in both stimulus categories for two different patch sizes. (**C**) t-values comparing house vs. face median activation in layer C1 across four different orientations and 8 scale bands (from smallest to largest receptive field). (**D**) t-values comparing house vs. face median activation in layer C2 for 8 different patch sizes. All p-values for the t-tests are less than 0.001.

that altering the strength of inhibitory connections in artificial neural networks can dramatically affect moment-to-moment dynamics and the number of different states that networks can sample (*Agrawal et al., 2018*). Similarly, decreasing GABA activity pharmacologically in healthy young rats and monkeys leads to a decrease in network dynamics and reduces the number of states visited by the cortical network (*Shew et al., 2011*). In humans, GABA levels are lower in older adults (*Chamberlain et al., 2021*; *Lalwani et al., 2019*; *Simmonite et al., 2019*; *Cassady et al., 2019*; *Porges et al., 2017a*; *Chalavi et al., 2018*; *Hermans et al., 2018*; *Gao et al., 2013*; *Cuypers et al., 2018*; *Levin et al., 2019*; *Porges et al., 2021*; *Porges et al., 2017b*) and in recent work, we found that overall cortical resting-state $SD_{BOLD}$ can be increased in older, poorer performing adults by GABA agonism (*Lalwani et al., 2021*). However, it remains unknown whether GABA provides a principled basis for how humans align brain signal variability to the complexity of visual input, a crucial step for understanding the mechanisms subserving task-relevant variability modulation in the human brain. Under the assumption that temporal dynamic range is required to distinctly represent external stimuli and respond to stimulus complexity (*Garrett et al., 2020*; *Młynarski and Hermundstad, 2018*), it is plausible that boosting GABA in individuals with lower baseline GABA levels should increase $\overrightarrow{\Delta}SD_{BOLD}$, while those with higher baseline levels of GABA would exhibit a decrease in $\overrightarrow{\Delta}SD_{BOLD}$, consistent with an inverted-U effect (*Northoff and Tumati, 2019*).

Using data from 58 younger (ages 18–25) and 77 older (ages 65–85) adults (see *Figure 1* for complete study design), we addressed these various questions using a combination of computational modeling, functional magnetic resonance imaging (fMRI), magnetic resonance spectroscopy (MRS), pharmacological intervention, and behavioral analyses. Specifically, we first utilized a computational model of the visual system (HMAX) to estimate the complexity of visual stimuli seen by participants during fMRI. To examine the GABAergic basis of variability modulation, we then (1) measured baseline visuo-cortical GABA levels using MRS, and (2) used lorazepam (a $GABA_A$ agonist) to causally increase GABA activity, while assessing visual $\overrightarrow{\Delta}SD_{BOLD}$ on and off drug. In a different subset of participants, we computed a latent visual discrimination behavioral score across four offline tasks. We tested whether: (1) neural variability scaled in response to stimulus complexity ($SD_{BOLD-HOUSES} > SD_{BOLD-FACES}$) (2) $\overrightarrow{\Delta}SD_{BOLD}$ was lower in older adults in comparison to young adults; (3) lower $\overrightarrow{\Delta}SD_{BOLD}$ was associated with lower visual GABA levels measured using MRS; (4) pharmacologically increasing GABA activity increased $\overrightarrow{\Delta}SD_{BOLD}$ in those with low GABA levels, and; (5) higher GABA and greater $\overrightarrow{\Delta}SD_{BOLD}$ were associated with better visuo-behavioral performance.

## Results

### HMAX analysis of visual Stimuli

We first estimated the complexity (or 'feature-richness') of our set of visual stimulus categories (faces and houses) by submitting them to a biologically inspired computational model of visual processing (HMAX; *Riesenhuber and Poggio, 1999*; *Serre et al., 2007a*; *Serre et al., 2007b*; *Serre, 2005*). To anticipate the results, we found that houses were more 'feature-rich' than faces in both C1 (corresponding to V1) and C2 layers (corresponding to extrastriate regions), as described in detail below, consistent with (*Garrett et al., 2020*; all t-values>10).

Physiological studies in non-human primates over recent decades have demonstrated that the receptive fields of neurons increase in both size and complexity as we move anteriorly along the ventral visual pathway. These insights are reflected in the biologically inspired, openly available HMAX feedforward model of visual recognition (code source: http://maxlab.neuro.georgetown.edu/hmax.html). The two earliest layers in this model (S1 and C1) correspond to neurons in primary visual cortex (V1) and the next two layers (S2 and C2) correspond to neurons in extrastriate visual areas (V2/V4). Following *Garrett et al., 2020* we used this model to objectively estimate the visual complexity of the two stimulus categories presented during our visual task (houses vs. faces). This model can also generate predictions about which cortical regions should be most sensitive to differences in stimulus complexity in our specific stimulus set. We focused our analyses on the C1 and C2 layers that aggregate the responses of cells in the S1 and S2 layers.

Layers in the first layer (S1) are modelled using Gabor functions with 16 different size filters (ranging from 7 to 37) corresponding to n x n pixel neighborhoods for four different orientations (Default: –45°,

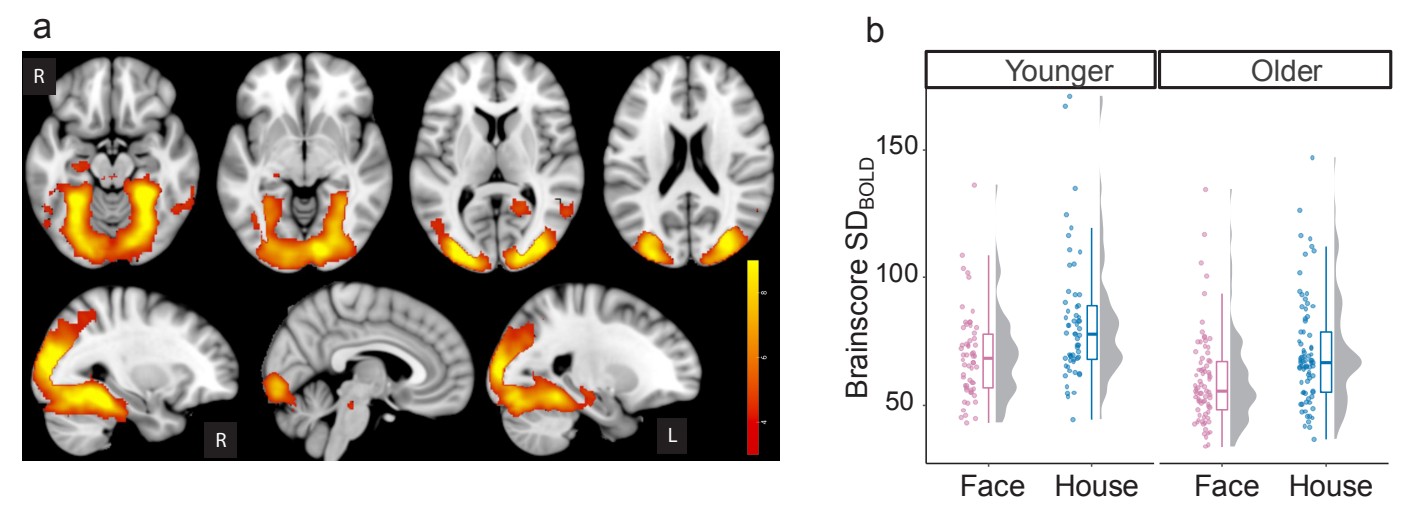

**Figure 2.** Effect of task-condition on SDBOLD. (**a**) Bootstrap ratios are thresholded at a value of ≥3.00, which approximates a 99% confidence interval and increase from red to yellow (no regions showed reliable decrease). (**b**) Primary and extended visual cortex show reliably higher variability (Cohen's $f$=0.46) during the house condition (in blue) than face condition (in pink) in both younger and older adults.

0°, 45°, 90°). For each orientation and filter size, a model was fit to each image in overlapping windows (50% overlap) resulting in a simple cell response map for all positions within the input image. At the next layer (complex cells in C1), the maximum activity over S1 units corresponding to each orientation is computed separately. Since 16 filter sizes were used, taking the maximum over neighboring pairs of filters results in eight 'scale bands'. The scale band index corresponds to the spatial neighborhood of S1 cells over which outputs are pooled. For each of the 8 scale-bands and 4 orientations, we calculated a median within-image C1 activation value for each image and then standardized them by computing z-scores. Using t-tests, we compared these within-image median values across the two stimulus categories. The results of these 8 (scales) × 4 (orientations) independent sample t-tests (*Figure 1a and c*) indicate that houses consistently produced a larger median C1 activation value than faces across all receptive field sizes.

In the third layer (S2), a template-matching approach is used. The receptive fields of the S2 cells correspond to a set of universal prototype templates derived from a library of naturalistic stimuli and their activation is computed based on the Euclidean distance between incoming C1 activity from all four orientations and the stored prototype for that S2 cell. For each prototype, an S2 map is computed across all positions at each of the 8 scale bands. The final layer (C2) then takes a global maximum over all scales and positions for each S2 map separately for different neighborhood (patch) sizes. We computed the median activation for each C2 neighborhood size separately and then standardized the results using z-scores. We then compared median activation by faces and houses using eight independent sample t-tests (one for each patch size). We found that house stimuli showed greater median activity compared with faces across different patch sizes (*Figure 1b and d*).

**Table 1.** Brain regions that exhibited a reliable association between task-condition and SD<sub>BOLD</sub>.

| Cluster number | MNI co-ordinates | | | Peak threshold (BSR) | Cluster size (in 2 mm voxels) | Cortical region label based on Harvard Oxford Cortical Atlas |
|---|---|---|---|---|---|---|
| | X | Y | Z | | | |
| 1 | 26 | −44 | −18 | 16.71 | 18565 | Temporal Occipital Fusiform Cortex Includes parts of Occipital Pole (AAL Label: Calcarine, Lingual), Lingual Gyrus, Temporal Fusiform Cortex, Occipital Fusiform Gyrus, posterior Parahippocampal Gyrus etc. |

## Visual stimulus complexity and variability modulation in younger and older adults

To examine whether brain signal variability is modulated in line with visual complexity, we computed the standard deviation of the BOLD signal ($SD_{BOLD}$) during each task condition (houses and faces) within each voxel and participant after extensively pre-processing the fMRI data (see Methods). Using Partial Least Squares (PLS), we found a single significant latent variable (p<0.001) revealing higher $SD_{BOLD}$ when viewing houses than faces ($SD_{BOLD-HOUSES} > SD_{BOLD-FACES}$), particularly in ventral visual cortex (See *Figure 2* for spatial extent peak and *Table 1* for further details). This task-condition effect on $SD_{BOLD}$ was significant in both younger (t(57) = 3.43, p<0.001, Cohen's d=0.45) and older adults (t(76) = 4.15, p<0.001, Cohen's d=0.47).

## GABA+ levels and modulation of variability

Raw GABA+/$H_2O$ levels (measured using MR spectroscopy) were significantly lower in older adults compared to younger adults (t (131.9)=–6.6, p=8e-10, Cohen's d=1.1). GABA +levels were also significantly lower in older adults compared to younger adults after correcting for tissue-composition differences (t(131.9)=–3.13, p=0.002, Cohen's d=0.53) after applying Harris et al.'s alpha tissue correction procedure (*Porges et al., 2017b*). These results suggest that the differences in tissue composition do not explain the observed lower GABA levels in older adults. Since, raw GABA+/H2O levels were highly correlated with alpha tissue composition-corrected GABA levels (r(131) = 0.88, p<2.2e-16), all results presented below were based on the raw GABA+/H2O levels. We further confirmed that the strongest task-condition effects on $SD_{BOLD}$ are present in the visual cortex and our GABA estimates are also in the task-relevant visual cortex. Thus, we restricted all further analyses within an anatomically defined task-relevant visual mask in all subjects (see Materials and methods) rather than investigate it in the whole-brain and in task irrelevant regions.

We investigated the role of Age and GABA levels in modulation of brain signal variability during visual task $\overrightarrow{\Delta}SD_{BOLD}$. We found a single significant latent variable (p=0.018) capturing a positive correlation between $\overrightarrow{\Delta}SD_{BOLD}$ and ventrovisual GABA levels and a negative correlation between $\overrightarrow{\Delta}SD_{BOLD}$ and age. As is typical when using multivariate PLS *Krishnan et al., 2011*, a *brainscore* was computed for each subject as the dot product between the resulting PLS model brain-pattern

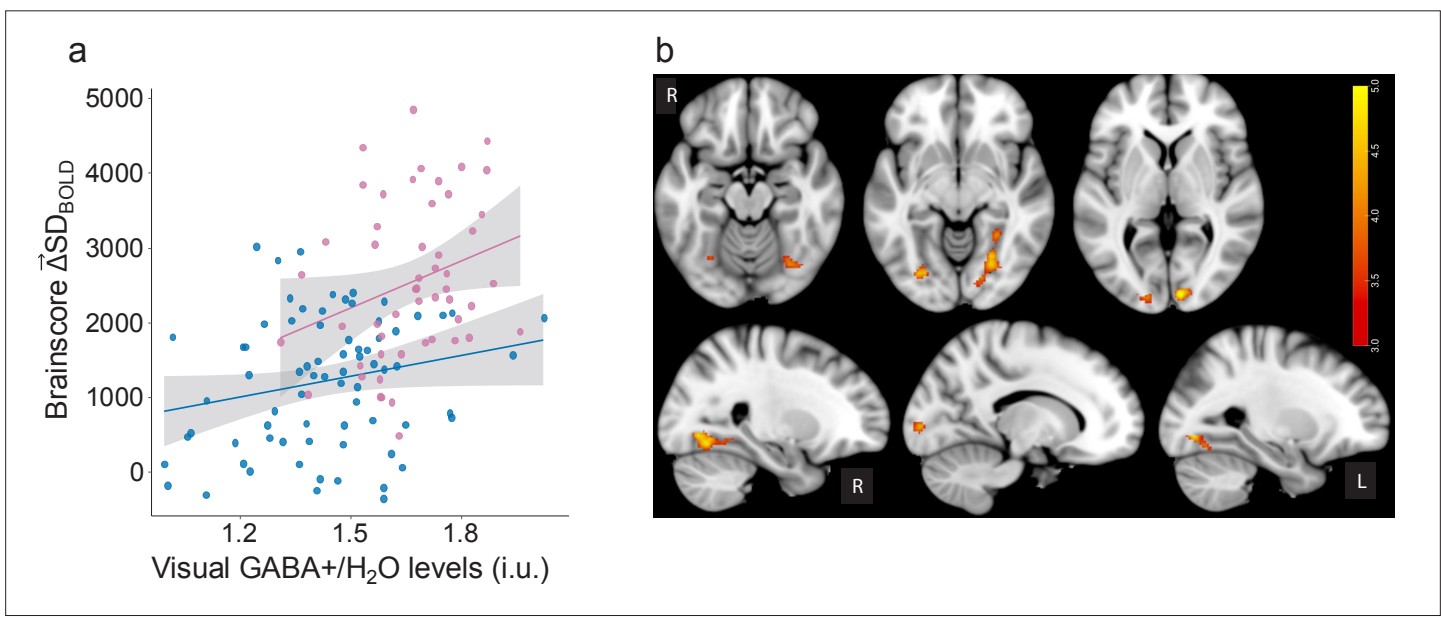

**Figure 3.** Higher GABA +levels are associated with greater $\overrightarrow{\Delta}SD_{BOLD}$. (**a**) Higher multivariate $\overrightarrow{\Delta}SD_{BOLD}$ is associated with younger adult age (Cohen's f=0.71) and with greater GABA +levels (Cohen's f=0.22) in both older (in blue) and younger adults (in pink). (**b**) Relationship between $\overrightarrow{\Delta}SD_{BOLD}$ and GABA levels is robust in primary visual cortex and fusiform gyrus. Bootstrap ratios are thresholded at a value of ≥3.00, which approximates a 99% confidence interval and increase from red to yellow.

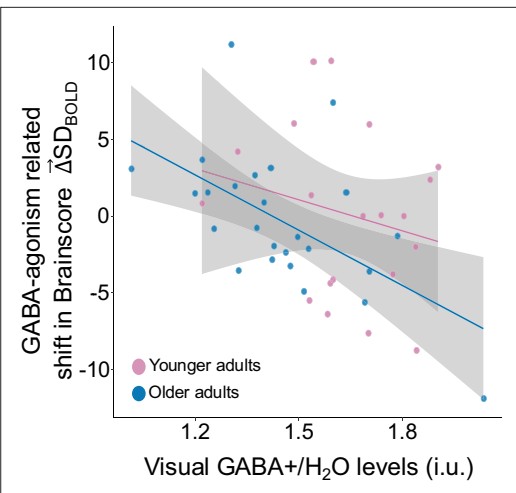

**Figure 4.** Drug-related shift in $\overrightarrow{\Delta}\mathrm{SD}_{\mathrm{BOLD}}$ is associated with baseline GABA +levels measured using MRS (off-drug). GABA agonism leads to an increase in $\overrightarrow{\Delta}\mathrm{SD}_{\mathrm{BOLD}}$ in participants with lower GABA levels, while those with higher GABA levels exhibit a decrease in $\overrightarrow{\Delta}\mathrm{SD}_{\mathrm{BOLD}}$ (Cohen's $f$=0.42) in both older adults (in blue) and younger adults (in pink).

(i.e. a vector of brain voxel weights; see Materials and methods) and $\overrightarrow{\Delta}\mathrm{SD}_{\mathrm{BOLD}}$ at each voxel. *Brain-scores* computed within a functional mask based on the $\mathrm{SD}_{\mathrm{BOLD}}$ Task-Condition model (consisting of all significant voxels from *Figure 2*) were highly correlated with those computed using the anatomical mask (r(133) = 0.95, p<2.2e-16). Thus, we used the brain pattern based on the entire anatomical mask for all further analysis which allows for better reproducibility in future studies.

Both age (F(1,120) = 62.8, p<1.3e-12, Cohen's $f$=0.71) and GABA (F(1,120) = 7.01, p=0.009, Cohen's $f$=0.22) also remained robustly associated with visual $\overrightarrow{\Delta}\mathrm{SD}_{\mathrm{BOLD}}$ after accounting for gray-matter volume differences. Higher GABA levels were similarly associated with greater $\overrightarrow{\Delta}\mathrm{SD}_{\mathrm{BOLD}}$ in both younger (r(50) = 0.34, p=0.01) and older adults (r(71) = 0.25, p=0.04) even after controlling for gray-matter volume differences. An Age x GABA interaction was not significant, F(1,120) = 1.07, p=0.3; (*Figure 3a*). The regions exhibiting a robust relationship (using 1000 bootstraps) between $\overrightarrow{\Delta}\mathrm{SD}_{\mathrm{BOLD}}$ and GABA levels were in the bilateral fusiform, calcarine and lingual cortex (see *Figure 3b* for spatial extent and *Table 2* for cluster details). These results suggest that individual differences in GABA levels play a role in how individuals modulate brain signal variability. Further, we found that tissue-composition corrected auditory GABA +levels (see *Lalwani et al., 2019* for details of voxel placement) were not associated with modulation of visual variability in the whole sample after controlling for age (F(1,120) = 0.72, p=0.4) or within older adults alone (r(74) = 0.01, p=0.9). Similarly, tissue-composition corrected motor GABA +levels (see *Cassady et al., 2019* for details of voxel placement) were not associated with modulation of visual variability in the whole sample after controlling for age (F(1,120) = 0.52, p=0.47) or within older adults alone (r(74) = 0.02, p=0.9), revealing that our GABA +effects were specific to visual cortex.

## Effect of GABA agonism on $\overrightarrow{\Delta}\mathrm{SD}_{\mathrm{BOLD}}$

To examine a more causal role of GABA, we administered lorazepam (a benzodiazepine known to potentiate the activity of GABA) in a subset of our participants (25 older and 20 younger adults). Masking by the same brain-pattern from the GABA–$\overrightarrow{\Delta}\mathrm{SD}_{\mathrm{BOLD}}$ model shown in *Figure 3*, we estimated the influence of drug on $\overrightarrow{\Delta}\mathrm{SD}_{\mathrm{BOLD}}$ ($\overrightarrow{\Delta}\mathrm{SD}_{\mathrm{BOLD-Drug}} - \overrightarrow{\Delta}\mathrm{SD}_{\mathrm{BOLD-Placebo}}$) and its relationship with baseline GABA +levels estimated during MRS-session (off-drug). Consistent with a GABAergic inverted-U account, we found that baseline GABA levels measured using MRS were negatively associated with the drug-related shift in $\overrightarrow{\Delta}\mathrm{SD}_{\mathrm{BOLD}}$ (F (1,37)=7.85, p=0.008, Cohen's $f$=0.42) across all subjects (see

**Table 2.** Brain regions that exhibited a reliable association between age, GABA and $\overrightarrow{\Delta}\mathrm{SD}_{\mathrm{BOLD}}$ .

| Cluster number | MNI co-ordinates | | | Peak threshold (BSR) | Cluster size (in 2 mm voxels) | Cortical region label based on Harvard Oxford Cortical Atlas |
|---|---|---|---|---|---|---|
| | X | Y | Z | | | |
| 1 | –28 | –78 | -6 | 5.35 | 118 | Occipital Fusiform Gyrus |
| 2 | 10 | –92 | 2 | 5.05 | 124 | Bilateral Occipital Pole (AAL Label: Calcarine and Lingual) |
| 3 | 24 | –70 | -8 | 4.69 | 199 | Occipital Fusiform Gyrus |

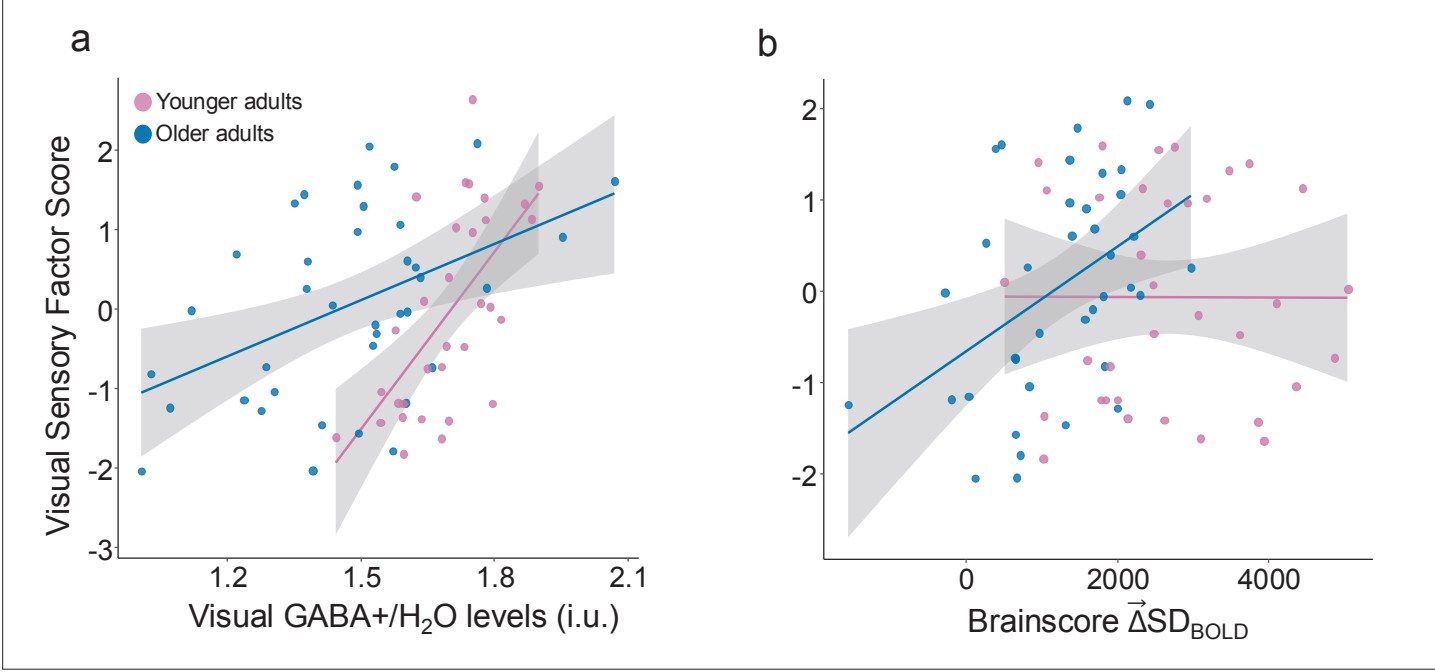

**Figure 5.** Visual sensory function, GABA +levels, and $\vec{\triangle}\mathrm{SD_{BOLD}}$. (**a**) Greater visual sensory scores are significantly associated with higher GABA levels in older adults (in blue) (rho(36) = 0.41) and younger adults (in pink) (rho(30) = 0.62). (**b**) Higher visual sensory scores were also significantly associated with $\vec{\triangle}\mathrm{SD_{BOLD}}$ in older adults (rho(36) = 0.41) but not in younger adults.

*Figure 4*), even after accounting for age, dosage, days between two sessions, order of sessions, and gray-matter volume differences. Age-Group did not interact with GABA level (F (1,37)=0.9, p=0.36).

Drug dosage was randomly assigned across participants independent of their baseline GABA levels, age or weight, such that 6 older and 7 younger participants received the 1 mg dosage. Baseline GABA levels across participants with 1 mg dosage did not differ significantly from those with 0.5 mg dosage in older (t(6) = –0.7, p=0.5) or younger adults (t(13) = –1.5, p=0.2). Moreover, the drug-related shift in $\vec{\triangle}\mathrm{SD_{BOLD}}$ did not differ significantly between participants who received 0.5 mg (Mean = 0.6) vs 1 mg (Mean = –2.0) dosage (t(20) = 1.6, p=0.13). These dosage effects are in the expected direction, but these findings should be interpreted with caution as only a small number (n=13) of participants received the 1 mg dosage of the drug and we are underpowered to detect the dosage-dependant effects on GABA-related shifts in $\vec{\triangle}\mathrm{SD_{BOLD}}$. Regardless, the relationship between baseline GABA levels and the drug-related shift in $\vec{\triangle}\mathrm{SD_{BOLD}}$ is significant even after controlling for the drug dosage in older adults (Pearson Correlation r(25) = –0.54, p=0.006).

Finally, the average effect of drug on $\vec{\triangle}\mathrm{SD_{BOLD}}$ was not significant (F (1,44)=0.04, p=0.85), unsurprisingly because adults with lower baseline GABA exhibited a drug-related increase in $\vec{\triangle}\mathrm{SD_{BOLD}}$, while those with higher baseline GABA showed a drug-related decrease.

## GABA+, $\vec{\triangle}\mathrm{SD_{BOLD}}$, and behavior

To investigate associations between behavior and both $\vec{\triangle}\mathrm{SD_{BOLD}}$ and GABA levels, a different subset of our participants (38 older and 36 younger adults) also completed four visual discrimination tasks: Buildings-in-noise, Faces-in-noise, Objects-in-noise and Scenes-in-noise. Using factor analysis, we computed a single latent factor score based on all these tasks (all loadings were positive; see Materials and methods).

We also observed a significant Age-Group x GABA level interaction on visual discrimination behavior, even after controlling for gray matter volume (F(1,65) = 7.13, p=0.01, Cohen's *f*=0.3). On examining the interaction further (See *Figure 5a*), we found that GABA levels were more strongly associated with better visual performance in younger adults (rho(30) = 0.62, p=0.0002, *Spearman's*

*correlation*) than in older adults (rho(36) = 0.37, p=0.022). Regardless, higher GABA levels were associated with better visual performance in both younger and older adults.

We also found that the *Brainscores,* computed from the previously presented GABA–$\overrightarrow{\Delta}SD_{BOLD}$ model, were significantly associated with visual performance in the older (rho(36) = 0.41, p=0.01), but not younger group (rho(33) = 0.002, p=0.99; see *Figure 5b*) and that the Age-Group x *Brainscore* interaction was significant (F(1,68) = 4.84, p=0.03, Cohen's *f*=0.23), even after controlling for gray matter volume. Thus, both greater $\overrightarrow{\Delta}SD_{BOLD}$ and higher GABA levels were associated with better visual discrimination performance in older adults.

Since, both $\overrightarrow{\Delta}SD_{BOLD}$ and GABA levels were associated with visual discrimination performance in older adults, we applied a hierarchical regression to compare a model explaining visual discrimination performance based on GABA levels alone (F(1,36) = 9.3, p=0.004, Adj. R²=0.18) to one including both $\overrightarrow{\Delta}SD_{BOLD}$ (F(1, 35)=10.42, p=0.003) and GABA levels (F(1,35) = 7.2, p=0.01; Total Model: F(2,35) = 8.8, p=0.0008, Adj. R²=0.3). We found that the model including both $\overrightarrow{\Delta}SD_{BOLD}$ and GABA explained significant variance over and above that explained by model based on GABA alone (F(1,35) = 6.8, p=0.01, Cohen's *f*=0.4) suggesting that $\overrightarrow{\Delta}SD_{BOLD}$ captures individual differences in visual sensory function beyond those related to differences in GABA levels in older adults.

## Results without excluding outliers

No outliers were determined using Cook's Distance (=4/sample size) in the model examining the role of task on $SD_{BOLD}$ or the role of baseline GABA in drug-related shifts in $\overrightarrow{\Delta}SD_{BOLD}$. In the linear model investigating the role of Age and GABA on $\overrightarrow{\Delta}SD_{BOLD}$, we found similar effects with and without excluding nine outliers determined by Cook's Distance. Both age (F(1,129) = 54.73, p<1.6e-11, Cohen's *f*=0.62) and GABA levels (F(1,129) = 7.03, p=0.009, Cohen's *f*=0.21) had a significant effect on this *Brainscore* even after accounting for gray-matter volume differences. Higher GABA levels were associated with greater $\overrightarrow{\Delta}SD_{BOLD}$ in both older and younger adults and the Age x GABA interaction was not significant (F(1,129) = 0.0004, p=0.98).

The model investigating the role of GABA in visual discrimination found that GABA had a significant effect on the visual performance latent score after controlling for age and gray matter volume (F(1,69) = 9.7, p=0.003, Cohen's *f*=0.34). However, the Age-Group x GABA interaction (F(1,69) = 0.74, p=0.39) was not significant. There were four young outliers with Cook's Distance greater than 0.054 (4/sample size). Nonetheless, the relationship between GABA and visual performance was significant in both age-groups, with or without outliers.

The *Brainscores* computed from the GABA–$\overrightarrow{\Delta}SD_{BOLD}$ model were significantly associated with the latent visual performance score without excluding the outlier (F(1,69) = 4.2, p=0.04, Cohen's *f*=0.21). The Age-Group x *Brainscore* interaction however showed a trend towards significance (F(1,69) = 3.6, p=0.06). One young outlier subject was determined as an outlier using Cook's Distance greater than 0.054 (4/sample size) and excluded from the results presented in the manuscript. Nonetheless, the correlation was significant only in older adults after controlling for age and gray matter volume (rho(37) = 0.41, p=0.01) but not in younger adults with (rho(34) = –0.01, p=0.95) and without the outlier (r(33) = –0.003, p=0.99).

## Results after controlling for visual acuity

Raw visual acuity (as measured by the NIH Visual Acuity task) was significantly lower in older adults (t(56.5)=3.85, p=0.0002), but it was not correlated with visual discrimination behavior score (r(72) = –0.03, p=0.78). This is not surprising as the discrimination task was performed with images presented at a comfortable distance and size to the participant. Moreover, controlling for visual acuity did not alter any of the reported effects. Both age (F(1,119) = 67.01, p<3.4e-13) and GABA (F(1,119) = 5.9, p=0.016) remained robustly associated with visual $\overrightarrow{\Delta}SD_{BOLD}$ after controlling for visual acuity and grey matter volume. Moreover, even after controlling for visual acuity and grey matter volume, both greater $\overrightarrow{\Delta}SD_{BOLD}$ (rho (36)=0.41, p=0.01) and higher GABA levels (rho (36)=0.35, p=0.03) were associated with better visual discrimination performance in older adults.

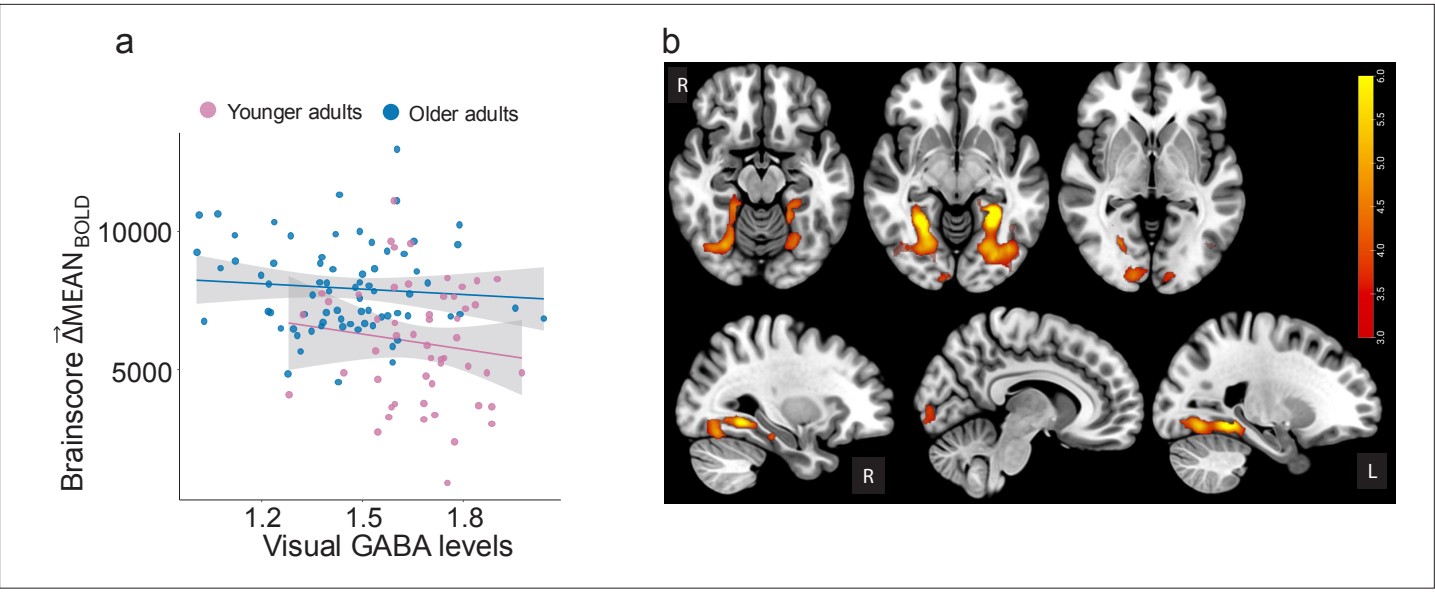

**Figure 6.** $\vec{\Delta}\text{MEAN}_{\text{BOLD}}$ and GABA +levels. (**a**) Higher multivariate $\vec{\Delta}\text{MEAN}_{\text{BOLD}}$ is associated with older adult age (F(1,129) = 36, p < 0.001) but not with GABA +levels in older (in blue) and younger adults (in pink). (**b**) Relationship between $\vec{\Delta}\text{MEAN}_{\text{BOLD}}$ and Age is robust in primary visual cortex and fusiform gyrus. Bootstrap ratios are thresholded at a value of ≥3.00, which approximates a 99% confidence interval and increase from red to yellow.

### MEAN$_{\text{BOLD}}$

Typically, SD$_{\text{BOLD}}$ effects are spatially and statistically distinct from MEAN$_{\text{BOLD}}$ effects in the literature (e.g. *Garrett et al., 2010*; *Garrett et al., 2011*; *Garrett et al., 2013a*; *Garrett et al., 2013b*; *Garrett et al., 2015*). To confirm this in the present sample, we examined $\vec{\Delta}\text{MEAN}_{\text{BOLD}}$ computed as the difference between MEAN$_{\text{BOLD-HOUSE}}$ and MEAN$_{\text{BOLD-FACE}}$ at each voxel. We found that while $\vec{\Delta}\text{MEAN}_{\text{BOLD}}$ was lower in younger adults (F(1,129) = 36, p<1.86e-08) in some similar regions as $\vec{\Delta}\text{SD}_{\text{BOLD}}$ (see *Figure 6* below), $\vec{\Delta}\text{MEAN}_{\text{BOLD}}$ was not related to GABA +levels overall (F(1,129) = 1.6, p=0.21), nor in young (r(55) = –0.02, p=0.88) or older adults (r(75) = –0.17, p=0.13). Moreover, $\vec{\Delta}\text{MEAN}_{\text{BOLD}}$ was also not related to visual-discrimination performance in older (r(36) = –0.24, p=0.14) or younger adults (r(34) = –0.06, p=0.7).

## Discussion

We report five main findings. First, brain signal variability in the visual cortex was modulated during various task conditions (greater when viewing more complex houses vs. less complex faces). Second, this modulation of variability ($\vec{\Delta}\text{SD}_{\text{BOLD}}$) and MRS-based GABA measures in the ventrovisual cortex were both significantly lower in older adults than in younger adults. Third, individual differences in $\vec{\Delta}\text{SD}_{\text{BOLD}}$ were significantly associated with GABA levels in the ventrovisual cortex. Fourth, participants with low GABA-levels tended to exhibit a drug-related increase in $\vec{\Delta}\text{SD}_{\text{BOLD}}$, while those with high GABA-levels tended to exhibit a drug-related decrease. Fifth, both higher GABA levels and greater $\vec{\Delta}\text{SD}_{\text{BOLD}}$ in the visual cortex were associated with better visual function in older adults. We discuss each of these findings in turn below.

### Visuo-cortical SD$_{\text{BOLD}}$ aligns with stimulus complexity and is lower in older adults

Consistent with prior research in older adults by *Garrett et al., 2020*, we replicated the finding that BOLD signal variability (SD$_{\text{BOLD}}$) is modulated based on the complexity of visual stimuli. SD$_{\text{BOLD}}$ in response to more 'feature-rich' house stimuli was greater than in response to simpler face stimuli in the visual cortex. Future research should investigate other types of stimuli (videos, objects, scenes

etc.) to quantify a parametric response in $SD_{BOLD}$ to feature richness in the visual cortex. Our current findings support the hypothesis that neural activity in the visual system should exhibit greater dynamic range in response to more complex input (*van Hateren, 1992*; *Waschke, 2023*). Indeed, greater variance in stimulus features has been argued to drive salience in the visual system (*Hermundstad et al., 2014*), which in turn could lead to an increase in resource allocation. It is hypothesized that the visual system (a) reduces resource allocation (narrowing neural dynamic range) when stimulus input is more reducible or less feature rich (here, faces), while it (b) upregulates dynamic range when stimulus input is more differentiated or feature rich (here, houses). Such upregulation of variability in response to the complexity of sensory input may reflect a well-adapted, dynamic neural network, akin to a 'well-adapted organism', as discussed by *Marzen and DeDeo, 2017*. Our finding that older adults were less able than younger adults to align neural variability to the complexity of visual input also converges with several previous studies showing that across-domain task-based variability modulation is increasingly muted with age (*Garrett et al., 2011*; *Garrett et al., 2010*; *Garrett et al., 2013a*). Older adults may indeed be less able to adapt to processing demands in their environment (*Garrett et al., 2013b*; *Waschke et al., 2021*).

## The GABAergic basis of neural variability modulation

In line with our assumptions, we found that lower baseline GABA levels in the ventrovisual cortex were significantly associated with lower $\overrightarrow{\Delta}SD_{BOLD}$ during visual processing in both young and older adults. Though both GABA levels and brain signal variability decline with age this relationship was significant even after controlling for age and tissue composition. This suggests that GABA levels play a role in individual differences in brain signal variability beyond the effects of age or tissue composition. Moreover, we found that this relationship was specific to visual cortex (i.e. auditory and motor GABA levels did not predict individual differences in visual $\overrightarrow{\Delta}SD_{BOLD}$). How does the brain's inhibitory activity (as measured by GABA levels here) play a role in modulation of brain signal variability? Computational modelling and animal research provide multiple in-roads.

Having sufficient inhibitory activity to offset excitatory activity has been found to be essential to allow artificial neural networks to operate near so-called criticality (*Shew et al., 2011*; *Poil et al., 2012*), allowing them to visit a variety of different network states, exhibiting higher dynamic range (*Agrawal et al., 2018*) and brain signal variability (*Lalwani et al., 2021*). Our finding that GABA (the brain's major inhibitory neurotransmitter) is related to brain signal variability is consistent with this finding. A flexible neural network which can sample a higher number of network states would be able represent different stimuli using a wider variety of different states - leading to more distinct representations of stimuli from each other. Consistent with this hypothesis, lower visual GABA levels in older adults have been associated with less differentiated responses to different visual stimuli in primates (*Leventhal et al., 2003*) and humans (*Chamberlain et al., 2021*). Our results extend these findings and suggest that lower GABA levels in older adults underlie a reduced ability to align neural variability to stimulus complexity, yielding impoverished stimulus differentiation in the older adult brain.

We proceeded to test the role of GABA in $\overrightarrow{\Delta}SD_{BOLD}$ more causally by administering lorazepam, a benzodiazepine known to potentiate GABA activity at $GABA_A$ receptors. Indeed, previous research in rodents noted that manipulating GABA levels in healthy rodents leads to a change in neural variability (*Shew et al., 2011*). We found in humans that the drug-related shift in $\overrightarrow{\Delta}SD_{BOLD}$ could be either positive or negative, while being negatively related to baseline GABA. Thus, boosting GABA activity with drug in participants with lower baseline GABA levels and low levels of $\overrightarrow{\Delta}SD_{BOLD}$ resulted in an increase in $\overrightarrow{\Delta}SD_{BOLD}$ (i.e. a positive change in $\overrightarrow{\Delta}SD_{BOLD}$ on drug compared to off drug). However, in participants with higher baseline GABA levels and higher $\overrightarrow{\Delta}SD_{BOLD}$, when GABA was increased (perhaps beyond optimal levels), participants experienced no-change or even a decrease in $\overrightarrow{\Delta}SD_{BOLD}$ on drug. These findings thus provide the first evidence in humans for an inverted-U account of how GABA may link to variability modulation.

Boosting low GABA levels in older adults helps increase $\overrightarrow{\Delta}SD_{BOLD}$, but why does increasing GABA levels lead to reduced $\overrightarrow{\Delta}SD_{BOLD}$ in those with higher baseline GABA levels? One explanation is that higher than optimal levels of inhibition can lead to dampening of the entire network. This reduced neuronal firing decreases the number of states the network can visit and decreases the dynamic range

of the network. Indeed, some anesthetics work by increasing GABA activity (for example propofol a general anesthetic modulates activity at $GABA_A$ receptors) and GABA is known for its sedative properties. Previous research showed that propofol leads to a steeper power spectral slope (a measure of the 'construction' of signal variance) in monkey ECoG recordings (*Gao et al., 2015*). Networks function optimally only when dynamics are stabilized by appropriate levels of inhibition. Thus, there is an inverted-U relationship between $\overrightarrow{\Delta}SD_{BOLD}$ and GABA that is similar to that observed with other neurotransmitters.

It is important to note that past work arguing for the presence of neurochemical inverted-U effects in humans has rarely estimated baseline neurochemical levels in specific brain regions prior to pharmacological manipulation, instead often relying on simple genotyping as a baseline estimate (e.g. COMT as a proxy for baseline dopamine *Bäckman et al., 2006*) or invasive PET-based imaging (*Cools et al., 2009*). MR spectroscopy may provide the easiest, regionally-specific baseline estimate of GABA for future tests of the GABAergic inverted-U.

## Higher GABA and variability modulation reflect better visual discrimination

If GABA and modulation of neural variability indeed help the brain better differentiate visual stimuli, then both should also manifest in better visual discrimination performance. Consistent with this hypothesis, we found that higher ventral visual GABA levels and higher $\overrightarrow{\Delta}SD_{BOLD}$ were both associated with better latent visual discrimination in-noise performance across different stimulus categories (buildings, faces, scenes, and objects). The GABA-behavior relationship was present in both age groups and is consistent with the hypothesis that older adults who maintain neurochemical levels comparable to those in younger adults experience fewer age-related impairments (*Nyberg et al., 2012*; *Bruffaerts et al., 2019*). Notably, $\overrightarrow{\Delta}MEAN_{BOLD}$ was not related to visual discrimination or GABA levels in young or older adults, further revealing $\overrightarrow{\Delta}SD_{BOLD}$ as a unique behaviorally and neurochemically relevant signature of brain function.

The role of GABA in visual performance is consistent with previous research which found that GABA levels in the primary visual cortex are associated with better orientation discrimination in neurons in corresponding early visual processing areas (*Kurcyus et al., 2018*) and with improved fluid processing ability (*Simmonite et al., 2019*). Individual differences in GABA levels in other brain areas are also linked with better sensory performance. For example, sensorimotor GABA levels predict motor function (*Cassady et al., 2019*; *Levin et al., 2019*), auditory GABA levels predict hearing loss (*Gao et al., 2015*; *Dobri and Ross, 2021*), and fronto-parietal GABA levels predict cognitive function across multiple domains assessed by higher MOCA scores (*Porges et al., 2017a*). The current work adds to this literature by demonstrating a behaviorally relevant and distinct role of GABA in the modulation of neural variability during visual processing.

Moreover, not only is greater task-based modulation of neural variability associated with better behavioral performance across a variety of general cognitive domains (*Garrett et al., 2010*; *Garrett et al., 2013a*; *Grady and Garrett, 2018*; *Waschke et al., 2021*; *Rieck et al., 2017*; *Rieck et al., 2022*), we show that the ability to align variability to stimulus complexity ($\overrightarrow{\Delta}SD_{BOLD}$) reflects visual discrimination directly, especially in older adults. Together, our results highlight the potential of targeting GABA activity in older adults with lower GABA levels to improve stimulus representations as characterized by $\overrightarrow{\Delta}SD_{BOLD}$, and in turn, visual processing.

We also found that within older adults, $\overrightarrow{\Delta}SD_{BOLD}$ explains variance in visual task performance beyond that explained by GABA. What might underlie these individual differences in older adults? Previous research has found that not only GABA, but dopamine levels also affect the modulation of variability during visual tasks (*Garrett et al., 2022*). Animal research also suggests that excitatory/inhibitory (E/I) *balance* plays a critical role in network flexibility (*Agrawal et al., 2018*). Thus, the level of excitatory neurochemicals (e.g. glutamate) might also be an important factor in age-related changes in sensory processing. Therefore, we hypothesize that along with GABA, other age-related neurochemical changes might also underlie individual differences in $\overrightarrow{\Delta}SD_{BOLD}$ and visual processing. A direct comparison of the role of dopamine, GABA, and glutamate would be ideal in future work.

The current research focused on understanding the role of GABA and $\overrightarrow{\Delta}\mathrm{SD}_{\mathrm{BOLD}}$ in visual processing. However, previous research has highlighted the role of higher $\overrightarrow{\Delta}\mathrm{SD}_{\mathrm{BOLD}}$ in better performance in other cognitive domains such as processing speed (*Garrett et al., 2013a*) and memory (*Garrett et al., 2022*; *Garrett et al., 2015*), while lower $\mathrm{SD}_{\mathrm{BOLD}}$ has been related to age-linked pathologies like Alzheimer's disease (*Millar et al., 2020*; *Zhang et al., 2020*). Separately, researchers have also found a link between dysfunction in GABAergic signaling and AD (*Calvo-Flores Guzmán et al., 2018*) as well as poor cognitive functions (*Porges et al., 2017a*) as described above. Investigating if age-related changes in GABA and $\overrightarrow{\Delta}\mathrm{SD}_{\mathrm{BOLD}}$ underlie these pathologies and cognitive decline might hold promise for developing new pharmacological targets to alleviate age-linked impairments.

## Limitations

One limitation of the current study is that participants who received the drug-manipulation did not complete the visual discrimination task, thus we could not directly assess how the drug-related change in $\mathrm{SD}_{\mathrm{BOLD}}$ impacted visual performance. Another important limitation is that the age comparisons were all cross-sectional. We therefore cannot rule out cohort effects which may have contributed to the observed differences between the young and older adults. MR spectroscopy also has limitations: (1) it only captures the baseline availability of GABA and (2) we are only able to measure the GABA +MM signal, an approximation of GABA using a standard MEGA-PRESS sequence that contains contamination from macromolecules (see Materials and methods). Although previous research has found GABA estimates using a special MM-suppression sequence and the GABA +MM signal from MEGA-PRESS to be correlated, there are regional and individual differences (see *Harris et al., 2015* for discussion on the relationship between GABA +and MM-suppressed GABA measures and how this relationship can vary across regions). Moreover, the MM signal, while often assumed to be functionally irrelevant, shows inter-individual differences and age-related differences that can limit the interpretations of our results.

## Conclusion

Our results suggest that GABA plays a key role in how humans of different ages modulate neural variability to adapt to the complexity of the visual world. The results suggest that interventions targeting inhibitory systems could potentially slow sensory and variability-based declines characteristic of older adults, a key goal for future work.

# Materials and methods

This data was collected as part of the Michigan Neural Distinctiveness (MiND) study. Here, we only describe the portions of the study that are relevant to this analysis. For details about the entire study protocol see *Gagnon et al., 2019*. The ethical approval for the study was granted by the Institutional Review Board of The University of Michigan (HUM00103117).

## Participants

We analyzed the data from 58 young (age 18–29 years) and 77 older (age 65 and above) adults who completed the entire MiND study before the start of the COVID-19 pandemic. All participants were recruited from Ann Arbor and the surrounding area, were right-handed, native English speakers, and had normal or corrected to normal vision. We screened out participants who scored 23 or lower on the Montreal Cognitive Assessment (MOCA) (*Carson et al., 2018*). All the sessions described below took place at the University of Michigan's Functional MRI Laboratory, Ann Arbor, Michigan. The following were subject inclusion and exclusion criteria:
   a. Study Inclusion Criteria

- Age 18–29 years or 65 years and older
- Right-handed
- Native English speaker
- Healthy (i.e. no debilitating conditions, mental illness, or head trauma)

   b. Study Exclusion Criteria

- Glaucoma
- Respiratory problems
- Benzodiazepine allergy
- Past or present chemotherapy
- Immune system disorder
- Kidney or liver disease
- Hearing problems or use of a hearing aid
- Color blindness
- Motor control problems
- Psychotropic medication
- Current depression or anxiety, or occurrence of depression/anxiety within 5 years
- Concussion with unconsciousness for 5 min or more
- Pregnancy or attempting to become pregnant
- More than four alcoholic drinks per week for women, more than six for men
- History of drug or alcohol abuse or addiction
- Weight greater than 250 pounds
- MRI incompatibility (claustrophobic, foreign metallic objects, pacemaker, etc)

## Power analysis

*Garrett et al., 2020* found a correlation of $r=0.47$ between modulation of visual variability and behavioral performance within older adults. In our convenience sample of those who completed the study before the pandemic and who completed the behavioral tasks (Subset 3), we had 80% power to detect a correlation of $r=0.47$ in each age-group alone.

## Session design

After completing an initial telephone screening interview and being determined eligible, all subjects participated in three sessions, each on a separate day. Session 1 only involved cognitive and behavioral testing, session 2 included a functional magnetic resonance imaging (fMRI) scan, and session 3 only involved a Magnetic Resonance Spectroscopy (MRS) scan (See *Figure 7* for summary).

## Subset determination

Each participant visited three times during the study. They completed behavior, fMRI, and MRS sessions on separate days. During the behavior sessions, participants completed several tasks including the NIH cognitive battery of tasks, Rivermead memory and Purdue Pegboard motor tasks. The details about all the tasks completed in the behavior session are described in *Gagnon et al., 2019*. Based on tasks and sessions completed participants were divided into three subsets.

The first 25 older adults and 20 younger adults enrolled in the study participated in two fMRI sessions (on and off drug) as well as a separate MRS session (described in detail below). They received either 0.5 mg or 1 mg of benzodiazepine during the on-drug fMRI session and the dosage was randomly assigned independent of their age-group, weight, or height. The average days between the two fMRI sessions was 18.72 days (Range 3–79 days), while that between placebo fMRI and spectroscopy session was 16.57 (Range 2–47). The drug-based component of the study was discontinued after the first-year review in February 2018.

The next subset (subset 2) of 14 older adults and 1 young adult thus only completed an off-drug fMRI and spectroscopy session with average days between two sessions of 10.5 (Range 2–28).

We later (June 2018) added more behavioral tests specifically targeting the visual perception in noise (described below in detail). Thus, a subset of participants (subset 3) who enrolled in the

**Table 3.** Participant distribution across three subsets.

| Subset | Completed dates | Number of participants | | Age | | Sex (female counts) | | Sessions completed | | | |
|---|---|---|---|---|---|---|---|---|---|---|---|
| | | Young | Older | Young | Older | Young | Older | fMRI placebo | fMRI drug | MRS | Vis Behav |
| 1 | February 2018 | 20 | 25 | 23.5 | 69.9 | 13 | 16 | ✓ | ✓ | ✓ | |
| 2 | June 2018 | 1 | 14 | 21 | 70.9 | 1 | 14 | ✓ | | ✓ | |
| 3 | March 2020 | 38 | 36 | 22.4 | 69.9 | 15 | 22 | ✓ | | ✓ | ✓ |

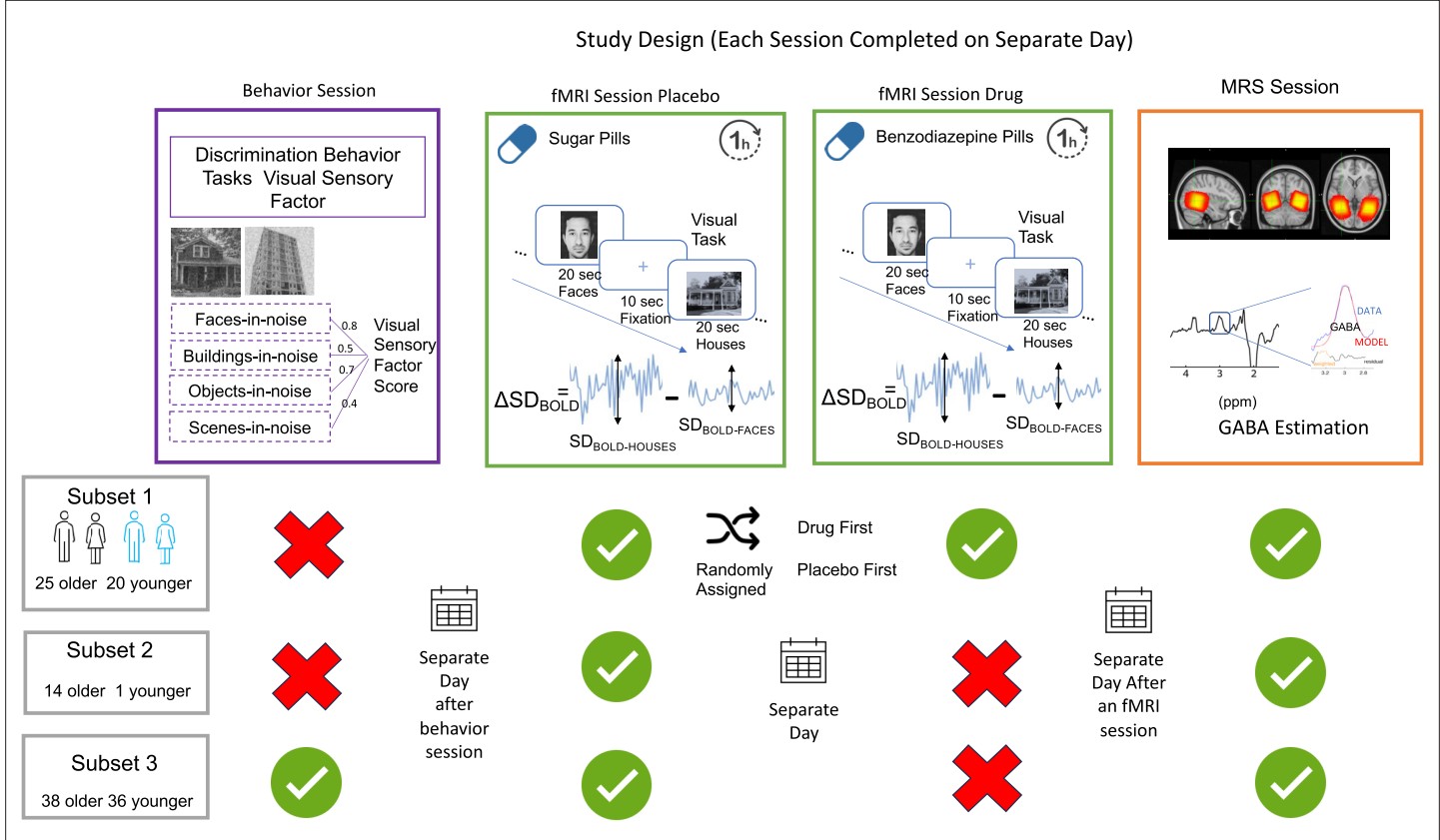

**Figure 7.** Session design and participant distribution. All participants underwent an fMRI and MRS scanning session on separate days. One subset of participants (25 older and 20 younger) received an additional on-drug fMRI scan on a separate day. The order of on-drug and off-drug fMRI sessions was randomized. A different subset (3) of participants (38 older and 36 younger) completed four visual discriminatory tasks on a separate day before fMRI testing. During the fMRI session participants completed a 6-min visual task with pseudorandomized 20 s blocks of passively viewed houses and faces interleaved with 10 s fixation blocks as shown in the green panel. Change in variability ($\overrightarrow{\Delta SD}_{BOLD}$) is computed at every voxel as the difference between $SD_{BOLD-HOUSES}$ and $SD_{BOLD-FACES}$. The orange panel shows the MRS voxel overlap across participants with brighter (yellow) indicating maximum overlap and red showing the least, an example spectrum obtained, and that raw GABA+/water is estimated by fitting a Gaussian model to compute the area under the curve of the 3ppm GABA peak.

study after June 2018 completed four visual tasks in noise that are relevant to the current study and described here along with the off-drug fMRI session and spectroscopy session on separate days (See *Table 3* for more details about participant split across sessions and source data file participant demographic details). For subset 3, the average number of days between behavior and fMRI session was 8.92 (Range 1–74), between fMRI and MRS session was 10.14 (Range 1–61), and between behavior and MRS was 19.05 (Range 2–78).

## Behavioral testing

A subset of participants (38 older and 36 younger adults) completed four visual tasks in noise, administered on a Dell laptop with a 15.6-inch screen using the Psychophysics Toolbox. Tasks were completed in a well-lit room where the laptop was placed at a fixed (same and measured for all participants) viewing distance of 21 inches. Participants were instructed to not lean forward or squint when completing the tasks. All tasks consisted of trials in which a 500ms fixation cross was followed by a black and white picture in dynamic Gaussian noise for 500ms followed by a response screen. The next trial began after the response. The order of the stimulus presentation was pseudorandomized but was the same across participants. Each task began with 4 practice trials with feedback and was followed by 50 scored trials without feedback. The tasks followed a staircase procedure – when a participant made three correct responses in a row, the level of noise was increased. Following an incorrect response, the level of noise was decreased. There were a total of 15 levels of Gaussian noise, and each task started

at the 5th level of noise. The dependent measure was the average level of noise presented for the last 40 trials. Thus, a higher score represents better performance.

a. Buildings in Noise (BIN)
The stimulus picture was either a house (50% of trials) or an apartment (50% of trials). Participants were asked to press '1' with their left index finger if they thought the picture was a house and '0' with their right index finger if they thought the picture was an apartment. Stimuli were from *Park et al., 2004*.

b. Faces in Noise (FIN)
The stimulus picture was either a male (50% of trials) or female (50% of trials) face. Participants pressed '1' if they thought the picture was a male face and '0' if they thought the picture was a female face. Stimuli were from *Gold et al., 1999*.

c. Objects in Noise (OIN)
The stimulus picture was either an office item, such as a stapler (50% of trials), or a food item, such as a hamburger (50% of trials). Participants pressed '1' for an office item and '0' for a food item. Object stimuli were taken from *Brady et al., 2008*.

d. Scenes in Noise (ScIN)
The stimulus picture was either an urban (50% of trials) or nature (50% of trial) scene. Participants pressed '1' if they thought the picture was an urban scene and '0' for a nature scene. Scene images were from *Zhou et al., 2018*.

Using the MATLAB function *factoran* we computed a single visual sensory factor based on these four tasks. The average level of noise presented for the last 40 trials for each of four task was used to do this. The loadings of the tasks on the factor were 0.3 (buildings), 0.8 (faces), 0.5 (objects) and 0.2 (scenes). Since, a higher score on each individual task represents better performance and each loading is positive a higher factor score also reflects better performance. Using principal component analysis (PCA) only one component had an eigenvalue >1, explaining 40% of the variance, and the corresponding weights were 0.53 (buildings), 0.8 (faces), 0.7 (objects), and 0.42 (scenes). The scores on this first component were correlated at ($r(72) = 0.88$, $p<2.2e-16$) with the factor scores, we thus present results based on factor score.

## fMRI session

MRI data were collected using a 3T General Electric Discovery Magnetic Resonance System with a volumetric quadrature bird cage head coil and two 32-channel receive arrays. A high-resolution T1-weighted structural image was collected using a 3D fast spoiled gradient echo (SPGR) BRAVO pulse sequence with the following parameters: TR = 12.2ms; TE = 5.2ms; TI = 500ms; flip angle = 15°; FOV = 256 × 256 mm; 156 axial slices with thickness = 1 mm and no spacing; acquisition time = 5 min and voxel size (1×1 × 1 mm).

The functional scan parameters were as follows: T2*-weighted images using a 2D Gradient Echo pulse sequence; Repetition Time (TR)=2000ms; Echo Time (TE)=30ms; flip angle = 90°; Field of View (FOV)=220 x 220 mm; 185 volumes; 43 axial slices; thickness = 3 mm, no spacing; collected in an interleaved bottom-up sequence; and voxel size (3×3 × 3 mm). The total acquisition time for the visual task scan was 6 min 10 s.

The task consisted of six 20 s blocks of images of male faces and six 20 s blocks of images of houses, presented in a pseudorandomized order and interleaved with twelve 10 s blocks of a fixation cross. Each block consisted of the stimulus presented for 500ms with an interstimulus interval (ISI) of 500ms. The block order was the same for all the participants. This was a passive viewing task but to ensure participant attention, we presented rare target trials once every minute (6 in total). The target trials were images of female faces for the face blocks, and images of apartment buildings for the house blocks. Participants were instructed to press a button with their right index finger every time they saw a target trial. Both age-groups performed well and similarly on correct target hits (Mean hits in young = 4.7 and Mean hits in older adults = 4.9). The fMRI sessions lasted approximately 45 min, including the acquisition of a structural image and other task-based (motor and auditory) and resting state functional data. The remainder of tasks completed in the fMRI session are described in detail in *Gagnon et al., 2019*, of which the motor and visual tasks were counterbalanced in their order of completion across participants and was always done after the structural, resting-state, and auditory scan.

## MRS session

Magnetic resonance spectroscopy (MRS) scanning was completed on a different day using the same MRI scanner. This session lasted 1.5 hr, during which we collected another T1-weighted structural image and MRS data. The T1-weighted image was obtained using the same parameters and sequence used during the fMRI session. The MRS data was obtained from 3cm x 3cm x 3 cm voxels placed in the left and right ventrovisual cortex. For each participant, we used their fMRI task sessions to compute an activation map for a face and house viewing vs. fixation contrast. These activation maps provided visual guidance to help with the voxel placement. Specifically, the fMRI technician used the activation maps to place the large MRS voxels so that they were centered roughly at the peak of the activation while not extending outside the brain. The voxels were rotated such that one of the edges was almost parallel and close to the cerebellum-cortical boundary minimizing overlap with dura, parietal, and angular regions. It was also rotated in the inferior-posterior plane to maximize coverage in the posterior occipital regions along with the fusiform and parahippocampal gyrus while minimizing overlap with the hippocampus. Note that these are large voxels, and the primary concern is to minimize overlap with non-cortical regions like ventricles, brainstem, dura, and cortex-cerebellum boundary that affect contamination and good shim (FWHM less than 15) while maximizing the coverage of region of interest. In this case the right and left ventrovisual regions were targeted and the voxel covered parts of fusiform, inferior occipital, calcarine, lingual, and parahippocampal gyrus. It was shifted away from the midline as much as possible without covering the dura. The voxel overlap across participants is shown in *Figure 7*. We used a MEGA-PRESS sequence with the following parameters to obtain MR spectra: TE = 68ms (TE1=15ms, TE2=53ms), TR = 1.8 sec, spec. width = 5 kHz, Frequency selective editing pulses (14ms) applied at 1.9ppm (ON) and 7.46 ppm (OFF), 256 transients (128 ON interleaved with 128 OFF) of 4096 data points (See *Table 4* for further details about the scan). For water referencing, eight transients of water unsuppressed data were collected at the end of the sequence, resulting in one reference frame where the NEX = 8 averaging is applied.

## Drug session

A subset of participants (20 young and 25 older adults) underwent two functional MRI sessions instead of one, one after taking a low dose benzodiazepine (lorazepam) and one after taking a placebo pill. The functional scanning parameters were identical to those described above. The pills were given approximately 1 hr before the session and the order of the sessions (on and off drug) was counterbalanced across participants. During the drug session, participants were administered a 0.5 or 1 mg oral dose of lorazepam (a benzodiazepine). The dosage was assigned randomly across participants independent of all other factors. Participants were not told which pill they received on which day (they were blind to the drug administration order) but the experimenter was not.

## fMRI data preprocessing

The fMRI data were preprocessed and analyzed using a combination of FMRIB Software Library (FSL), SPM12 and MATLAB-based scripts. The first 5 volumes of each scan were discarded. Heart rate was collected via a pulse oximeter placed on the left middle finger and the data was physio corrected during preprocessing. We performed first-level preprocessing using FSL-FEAT *Woolrich et al., 2001* with default parameters for motion correction and normalization. Spatial smoothing with a 7 mm full-width at half maximum Gaussian kernel was performed. We used the SPM12 function 'spm detrend' to remove linear, quadratic and cubic trends in the time series and also applied a Butterworth filter (0.01–0.1 Hz). We then ran FSL MELODIC to perform Independent Component Analysis (ICA). Three separate raters identified noise components through manual visual inspection *Kelly et al., 2010*. These components reflected noise related to sinus activity, vascular and ventricle activations, and motion. We then removed the components identified as noise by at least two of the three raters using the FSL regfilt function. Subsequently, we performed linear registration of the functional and anatomical images of each participant and the MNI152 template using the FSL FLIRT function.

## Quantification of brain signal variability (SD$_{BOLD}$)

After preprocessing the fMRI data, we normalized all blocks for each condition (houses and faces separately) such that the overall 4D mean across brain and the block was the same. Then, for each voxel, we subtracted the block mean and concatenated the time-series across all blocks for that

**Table 4.** Minimum reporting standard table for MRS session.

| 1. Hardware | |
|---|---|
| a. Field Strength [T] | 3T |
| b. Manufacturer | GE |
| c. Model (software version if available) | Discovery MR750 DV26R2 |
| d. RF coils: nuclei (transmit/receive), number of channels, type, body part | 8-channel head coil |
| e. Additional hardware | NA |
| **2. Acquisition** | |
| a. Pulse sequence | MEGA-PRESS **Mescher et al., 1998** |
| b. Volume of Interest (VOI) locations | Seven VOI collected:<br><br>• Left primary visual<br>• Left ventral visual<br>• Right ventral visual<br>• Left auditory cortex<br>• Right auditory cortex<br>• Left somatosensory cortex<br>• Right sensorimotor cortex<br><br>See heatmaps with VOI locations across all participants for relevant visual cortex in **Figure 7** |
| c. Nominal VOI size [cm3, mm3] | 30x30 x 30 mm$^3$ |
| d. Repetition time ($T_R$), echo time ($T_E$) [ms, s] | $T_R$ = 1800ms, $T_E$ = 68ms<br>($T_E$1 = 15ms, $T_E^2$ = 53ms) |
| e. Total number of excitations of or acquisitions per spectrum<br>In time series for kinetic studies<br>i. Number of averaged spectra (NA) per time point<br>ii. Averaging method (eg block-wise or moving average)<br>iii. Total number of spectra (acquired/in time series) | 256 (128 ON interleaved with 128 OFF) |
| f. Additional sequence parameters (spectral width in Hz, number of spectral points, frequency offsets) If STEAM: mixing time ($T_M$) If MRSI: 2D or 3D, FOX in all directions, matrix size, acceleration factors, sampling method | Spectral width = 5 kHz<br>Data points = 4,096<br>Frequency selective editing pulses applied at:<br>ON pulse = 1.9 ppm<br>OFF pulse = 7.46 ppm |
| g. Water suppression method | CHESS |
| h. Shimming method, reference peak, and thresholds for "acceptance of shim" chosen | Auto-shim, if linewidth >15 then voxel placement adjusted and shim redone. |
| i. Triggering or motion correction method (respiratory, peripheral, cardiac triggering, incl. device used and delays) | None |
| **3. Data analysis methods and outputs** | |
| a. Analysis software | Gannet 3.1 in MATLAB |
| b. Processing steps deviating from quoted reference or product | No deviations – standard steps were used. |
| c. Output measure (eg absolute concentration, institutional units, ratio), processing steps deviating from quote reference or product | Gannet outputs:<br>- Ratio to Cr (GABA/Cr)<br>- Ratio to NAA (GABA/NAA)<br>- Institutional units relative to water, uncorrected for VOI tissue composition (GABA/H$_2$O)<br>- Institutional units relative to water, corrected for VOI CSF content (GABA/H$_2$O CSF)<br>- Institutional units relative to water, corrected for tissue composition of VOI (GABA/H$_2$O TC)<br>- Institutional units relative to water, corrected for tissue composition of VOI, and alpha of.5 applied to white matter % (GABA/H$_2$O ATC)<br>- Institutional units relative to water, corrected for tissue composition of VOI, and alpha of.5 applied to white matter %, and normalised to average tissue composition of VOI in young adults (GABA/H$_2$O GATC) |

*Table 4 continued*

| 3. Data analysis methods and outputs | |
| --- | --- |
| d. Quantification references and assumptions, fitting model assumption | |
| **4. Data quality** | |
| a. Reported variables | GABA +fit error |
| b. Data exclusion criteria | - Spectra were visually inspected for GABA peaks and model fits.<br>- Gannet derived estimates were included in analyses if GABA +fit error was below 15% |
| c. Quality measures of post-processing model fitting (eg CRLB, goodness of fit, SD of residual) | **Gannet:** Fit Error % |
| **5. Average Spectra** | |

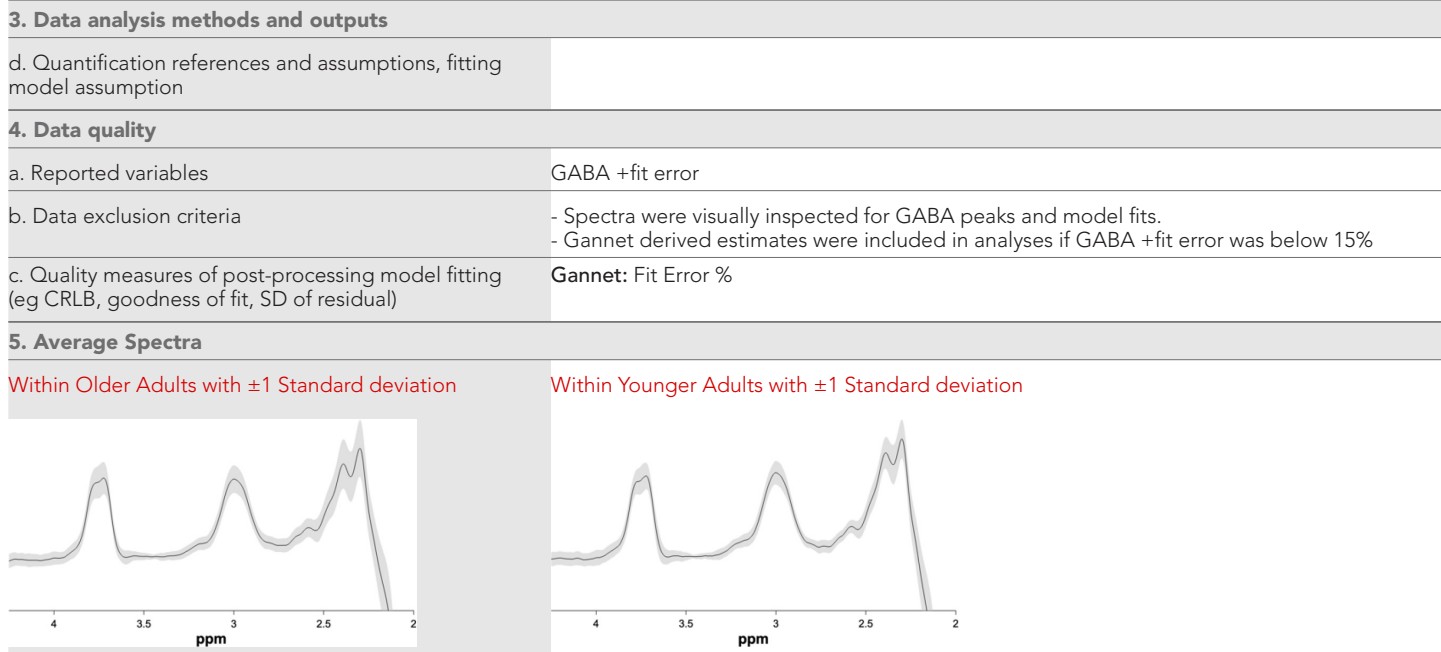

Within Older Adults with ±1 Standard deviation

Within Younger Adults with ±1 Standard deviation

condition. The standard deviation in this time-series during each of the conditions (faces and houses) was computed at each voxel for each participant (similar to previous work *Garrett et al., 2020*). Modulation of variability ($\vec{\Delta}\mathrm{SD}_{\mathrm{BOLD}}$) was computed as the difference between the two conditions ($\mathrm{SD}_{\mathrm{BOLD\text{-}HOUSES}} - \mathrm{SD}_{\mathrm{BOLD\text{-}FACES}}$) at each of the voxels in a grey matter volume mask. Similarly, $\vec{\Delta}\mathrm{SD}_{\mathrm{BOLD}}$ on drug was computed for the subset of participants who were administered lorazepam.

## Quantification of GABA levels

We used Gannet 3.0 (*Edden et al., 2014*), a MATLAB based toolbox, to estimate GABA+/Water levels based on the MEGA-PRESS difference spectra in each of the MRS voxels. All the time-domain data were phase corrected and frequency corrected using spectral registration and filtered with 3 Hz exponential line broadening and zero-filled by a factor of 16. The GABA levels were scaled to water and expressed in institutional units by Gannet. Gannet quantifies GABA levels by fitting a five-parameter Gaussian model to the MR spectrum between 2.19 and 3.55 ppm while the water peak is modelled using a Gaussian-Lorentzian function. The MEGA-PRESS editing scheme also results in excitation of coedited macromolecules (MM), which can contribute up to 45% to the edited signal around 3ppm overlapping with the GABA peak. Thus, all GABA values are reported as GABA+ (i.e. GABA +MM) in the present study.

Gannet's integrated voxel-to-image co-registration procedure produces a binary mask of the MRS voxel. Using an SPM-based segmentation function, Gannet estimates the tissue composition (voxel fractions containing Cerebrospinal Fluid (CSF), Grey Matter (GM) and White Matter (WM)). Gannet then estimates a tissue-corrected GABA +value that accounts for the fraction of grey matter, white matter, and CSF in each MRS voxel as well as the differential relaxation constants and water visibility in the different tissue types (*Harris et al., 2015*). Based on a quality control check of the spectra (flagged if the value was more than 3 standard deviations from mean, the fit error was greater than 10% for water, or the CSF fraction was more than 20%), we flagged and discarded three right and seven left GABA values. GABA measures between right and left ventrovisual voxel were correlated at $r(134) = 0.57$ ($p<0.0001$). Thus, we computed an average GABA +measure for each participant. One young participant had both right and left ventrovisual GABA values flagged and was excluded from the analysis.

## Statistical analysis

To estimate the effects of task-condition on $SD_{BOLD}$ and the effect of age and GABA on $\overrightarrow{\Delta}SD_{BOLD}$ we employed multivariate partial least squares (PLS) analyses (***McIntosh et al., 1996***).

1) To estimate the effects of task-condition on $SD_{BOLD}$ we used a task-PLS. In this method, first a between-subject covariance (COV) matrix is computed between house and face conditions and each voxel's $SD_{BOLD}$. Then a left singular vector of experimental condition weights (U) is estimated, along with a right singular vector of brain voxel weights (V) and a diagonal matrix of singular values (S). Significance of the detected relations is assessed using 1000 permutation tests of the singular value corresponding to each latent variable (LV). This resulted in two LVs of which only one was significant and represented greater variability during the house condition than during the face condition. *Brainscore* was computed separately for each condition as the dot product of brain voxel weights and each subjects' $SD_{BOLD-HOUSES}$ and $SD_{BOLD-FACES}$.

2) We employed rank-based Behavior-PLS for investigating the effects of age and GABA levels on $\overrightarrow{\Delta}SD_{BOLD}$ in a visual anatomical mask within our sample of 134 subjects (after excluding the one subject whose GABA estimates were flagged). The visual anatomical mask is a broad mask that contained the occipital pole, lingual gyrus, inferior division of lateral occipital cortex, the temporooccipital part of the inferior and middle temporal gyrus, temporal occipital fusiform cortex, occipital fusiform gyrus, and parahippocampal gyrus from the Harvard-Oxford atlas in FSL. A between-subject correlation matrix (CORR) was computed between each voxel's $\overrightarrow{\Delta}SD_{BOLD}$ (i.e. $SD_{BOLD-HOUSES} - SD_{BOLD-FACES}$) within this mask and both – (1) MRS-based average ventrovisual GABA measures and (2) self-reported age (in years). Then, CORR was decomposed using singular value decomposition (SVD). This decomposition produced a matrix of behavior weights (U), a matrix of brain voxel weights (V), and a diagonal matrix of singular values (S). A single significant latent variable captured the activity pattern depicting the brain regions that show the strongest relationship of $\overrightarrow{\Delta}SD_{BOLD}$ with both age and GABA. The behavior weights (U) of this LV suggest lower GABA levels and greater age were associated with lower $\overrightarrow{\Delta}SD_{BOLD}$. We obtained a summary measure of each participant's expression of this LV's spatial pattern (a within-person "*Brainscore*") by taking a dot product of the brain weights (V) with $\overrightarrow{\Delta}SD_{BOLD}$ on placebo (and on drug for the subset that received the manipulation). This *Brainscore* was also used for investigating the role of $\overrightarrow{\Delta}SD_{BOLD}$ on visual discrimination task.

In both models, we used a bootstrapping procedure (1000 bootstrapped resamples) to reveal the robustness of voxel weights. By dividing each voxel's weight by its bootstrapped standard error, we obtained 'bootstrap ratios' (BSRs) as estimates of robustness. We thresholded BSRs at values of ±3.00 (~99.9% confidence interval). Statistical analyses were conducted using R (***R Development Core Team, 2013***). Figures were plotted using the ggplot2 package (***Wickham, 2016***) and the lme4 package (***Bates et al., 2007***) was used to perform the linear mixed effects analysis. We used a Cook's Distance greater than 4/sample size to flag and remove outliers from each of mixed effects models reported in the results.

## Acknowledgements

HHS | NIH | National Institute on Aging (NIA): Poortata Lalwani, AG068517; HHS | NIH | National Institute on Aging (NIA): Thad Polk, AG050523 The funders had no role in study design, data colection and interpretation, or the decision to submit the work for publication.

# Additional information

### Funding

| Funder | Grant reference number | Author |
|---|---|---|
| National Institute on Aging | AG068517 | Poortata Lalwani |
| National Institute on Aging | AG050523 | Thad Polk |

The funders had no role in study design, data collection and interpretation, or the decision to submit the work for publication.

### Author contributions

Poortata Lalwani, Conceptualization, Formal analysis, Writing – original draft; Thad Polk, Supervision, Funding acquisition, Writing - review and editing; Douglas D Garrett, Conceptualization, Supervision, Methodology, Writing - review and editing

### Author ORCIDs

Poortata Lalwani (ID) https://orcid.org/0000-0003-3022-3152
Douglas D Garrett (ID) https://orcid.org/0000-0002-0629-7672

### Ethics

Informed consent was obtained from all participants in the study. The ethical approval for the study was granted by the Institutional Review Board of The University of Michigan (HUM00103117).

### Decision letter and Author response

Decision letter https://doi.org/10.7554/eLife.83865.sa1
Author response https://doi.org/10.7554/eLife.83865.sa2

---

# Additional files

### Supplementary files

MDAR checklist

Supplementary file 1. De-identified processed data for all participants.

### Data availability

Since, the raw MRI, MRS and fMRI data used in the current manuscript are collected as a part of a larger (MIND) study they will be made publicly available only on completion of the main study (estimated: 2027 based on the umbrella policies applicable for the larger study). Before that, data can be requested via email correspondence for research purposes. For data transparency and reproducibility, an Excel-sheet with all the de-identified values to reproduce all figures and results has been included as *Supplementary file 1*. The processing Matlab code is based on previous work and available on GitHub (*umich-tpolk-lab, 2021*).

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
