## [Editor Report]

This study by Lalwani et al. combines multiple complementary neuroscientific methods to understand the neural response to visual stimulus complexity in the human brain across lifespan. Lalwani and colleagues provide important and convincing evidence, drawing from appropriate and validated methodology. The manuscript has been comprehensively revised and now includes key control analyses, improved content throughout and clarifications. The study is of broad interest to neuroscientists and biologists interested in aging and sensory processing.

---

## [Decision Letter]

**Decision letter after peer review:**

Thank you for submitting your article "The modulation of neural variability reflects aging, GABA and behavior" for consideration by *eLife*. Your article has been reviewed by 3 peer reviewers, including I. Betina Ip as the Reviewing Editor and Reviewer #1, and the evaluation has been overseen by Senior Editor. The following individuals involved in the review of your submission have agreed to reveal their identity: Ru-Yuan Zhang (Reviewer #3).

Essential revisions (for the authors):

1. Greater effort can be made to make this paper of broader appeal. In the introduction unpack more the specific approach and relate it to more common measures of the hemodynamic response.

2. The MRS element appears generally well conducted, but more information should be provided. (a) What were the shim values for the voxels in the different groups? (b) What was the quality control (line 217)? (c) How were voxels positioned beyond centering over peak activation? Figure 1 suggests voxels were rotated in at least two planes based on anatomy. This information is missing from the manuscript. (d) A separate water peak estimate was conducted as described in the supplements, but it is not clear whether an additional PRESS scan was acquired. Please clarify. (e) Move the supplemental information about tissue-corrected GABA measurements to the main text. Otherwise, lines 212 to 221 make it somewhat ambiguous as to which GABA measurement was used in subsequent analyses.

3. It isn't clear to me how the measure of BOLDsd relates to conventional glm-approaches that isolate regions with stimulus preference. As far as the data shows, the BOLDsd approach does not isolate object-selective regions. Please lay out in more detail the rationale for using BOLDsd instead of conventional GLM contrast in the introduction; report the result of the glm contrast faces>houses in percentage BOLD-signal change and discuss their results in light of the different analyses approaches. This would also help broaden the accessibility of your article.

4. Baseline GABA+ and the task-condition effect on BOLD variability differed between age groups. Combining the two groups to then test for a relationship between GABA and the task-condition effect on BOLD variability can suggest a relationship that may be explained by age alone. The analyses should be repeated with age as a nuisance regressor. Along these lines, Figure 3 shows separate trend lines for the two groups, but the effect was reported across both groups in the text. The figure implies that both groups independently exhibit a relationship, but that is not the case.

5. Lines 173-174. The lorazepam manipulation is an important feature of the study. However, it is not clear why the two different dosages (.5 or 1 mg) were used and how many participants from each age group received which dosages. Also, were these dosages administered equally across individuals with different baseline GABA levels? Additional analyses should be reported to address whether the different dosages had different effects on BOLD variability across age groups and baseline GABA levels.

6. I am concerned about the definition of neural variability. In electrophysiological studies, neural variability can be defined as Poisson-like spike count variability. In the fMRI world, however, there is no consensus on what neural variability is. There are at least three definitions. One is the variability (e.g., std) of the voxel response time series as used here and in the resting fMRI world. The second is to regress out the stimulus-evoked activation and only calculate the std of residuals (e.g., background variability). The third is to calculate variability of trial-by-trial variability of β estimates of general linear modeling. It currently remains unclear the relations between these three types of variability and other factors. It also remains unclear the links between neuronal variability and voxel variability. I don't think the computational principles discovered in neuronal variability also apply to voxel responses. I hope the authors can acknowledge their differences and discuss their differences.

7. If I understand it correctly, the positive relationship between stimulus complexity and voxel variability has been found in the author's previous work. Thus, the claims in the abstract in lines 14-15, and section 1 in results are exaggerated. The results simply replicate the findings in the previous work. This should be clearly stated.

8. It is difficult for me to comprehend the U-shaped account of baseline GABA and shift in deltaSDBOLD. If deltaSDBOLD per se is good, as evidenced by the positive relationship between brainscore and visual sensitivity as shown in Figure 5b and the discussion in lines 432-440, why the brain should decrease deltaSDBOLD ?? or did I miss something? I understand that "average is good, outliers are bad". But a more detailed theory is needed to account for such effects. Relatedly, can you show the relationship between the shift of deltaSDBOLD (i.e., the δ of deltaSDBOLD) and visual performance?

*Reviewer #1 (Recommendations for the authors):*

Greater effort can be made to make this paper of broader appeal. In the introduction unpack more the specific approach and relate it to more common measures of the hemodynamic response. The MRS element appears generally well conducted, although basic information on data acquisition and spectral quality is missing.

– 81. What determined which participants were in which of the 3 subsets? Please name the decision criteria.

– What determines the cut-off point for old and young participants?

– 82. Participants – should include sex. Were participants screened for typical MRS exclusion criteria?

– 89. Does the power analysis refer to a specific subset? It could be subset 3 (38 older and 36 younger) but I am not sure. Please state the sample size and the subset.

– 95. I found the text describing the protocol and Figure 1 confusing. All three subsets took part in 3 sessions, but only subset 3 took part in the visual discrimination task. Subset 1 would have taken part in 2 imaging sessions, 1 for the drug and 1 for the placebo. Were those two imaging sessions identical (fMRI + MRS)? This needs to be laid out in detail.

– What exactly were the cognitive and behavioural testing components for Subset 1 and 2? Also, please note the average difference in days between sessions 1-3 for cohorts.

– Figure 1. I like the use of icons and would suggest an alternative icon to a person with a walking stick to represent people above 77 yo – the alternative could be the same icon as the younger population with a different shading/color.

– 104. Not sufficient information to understand what was done here. More information is needed on the testing, such as whether there was a head and chin rest, what was the viewing distance, whether the room was illuminated or dark etc.

– 143. What was the spatial resolution of the EPI scan?

– 146. Was the whole fMRI session 6 min- how long did the whole session last and was the fMRI scan always in the same order?

– 154. What was the average hit rate for the target trials, and were there differences across age groups?

– 156. MRS session – how were the voxels placed? Did you run another visual task scan to provide a map in the MRS session? The text suggests this is the case, but it is not explicitly mentioned. If you didn't use a localiser map from the same session, please describe how the "person-specific task-activations" were used to guide voxel placement.

– What were participants doing during the MRS scan and how many spectral averages were collected? Please include the document "minimum reporting standards" for MRS.

– 168. How was this subset chosen, and how were the drug and placebo randomised? Was the experimenter blind to the randomisation?

– 173. Why were there two dosages and were low vs high doses balanced?

– 183. 7 mm is not the default smoothing in FEAT.

– 192. Methodological detail missing for quantification of BOLDsd. Please refrain from referring to the previous paper. All information required for replicating the study should be in this current paper.

– 208. cite paper for MM contamination in the occipital cortex by Harris et al., 2015.

– 217. 'quality control check' define and report values.

– 263. It isn't clear to me how the measure of BOLDsd relates to conventional glm-approaches that isolate regions with stimulus preference. As far as the data shows, the BOLDsd approach does not isolate object-selective regions. Please lay out in more detail the rationale for using BOLDsd instead of conventional GLM contrast in the introduction; report the result of the glm contrast faces>houses in percentage BOLD-signal change and discuss their results in light of the different analyses approaches. This would also help broaden the accessibility of your article.

– 288-289 This sentence struck me as odd and makes the analysis sound post hoc.

– 314: I'm not sure how the data in Figure 4 was pulled together, this comes back to Figure 1 where more detail is needed on what exactly different people were tested on, and in what order. Is the baseline GABA from one of the on/off-drug sessions, or is it separate? I am guessing that the y-axis is the difference between placebo and drug in fMRI, but that the x-axis is from a separate off-drug MRS session. Whatever the case needs to be clarified.

– GABA levels should be noted with units on the axis GABA+/water IU.

– 319: Why is the change due to drug consistent with an inverted-U account if the change simply describes a negative relationship of BOLDsd to baseline GABA?

*Reviewer #2 (Recommendations for the authors):*

1. Baseline GABA+ and the task-condition effect on BOLD variability differed between age groups. Combining the two groups to then test for a relationship between GABA and the task-condition effect on BOLD variability can suggest a relationship that may be explained by age alone. The analyses should be repeated with age as a nuisance regressor. Along these lines, Figure 3 shows separate trend lines for the two groups, but the effect was reported across both groups in the text. The figure implies that both groups independently exhibit a relationship, but that is not the case.

2. Further information should be included about the MRS imaging approach. (1) What were the shim values for the voxels in the different groups? (2) What was the quality control (line 217)? (3) How were voxels positioned beyond centering over peak activation? Figure 1 suggests voxels were rotated in at least two planes based on anatomy. This information is missing from the manuscript. (4) A separate water peak estimate was conducted as described in the supplements, but it is not clear whether an additional PRESS scan was acquired. Please clarify. (5) Move the supplemental information about tissue-corrected GABA measurements to the main text. Otherwise, lines 212 to 221 make it somewhat ambiguous as to which GABA measurement was used in subsequent analyses.

3. Lines 173-174. The lorazepam manipulation is an important feature of the study. However, it is not clear why the two different dosages (.5 or 1 mg) were used and how many participants from each age group received which dosages. Also, were these dosages administered equally across individuals with different baseline GABA levels? Additional analyses should be reported to address whether the different dosages had different effects on BOLD variability across age groups and baseline GABA levels.

4. Lines 158-160. Please include the scanning parameters for the T1-weighted MRI in the Methods section even if they are reported in a previous publication.

5. Lines 308-310. Were the bilateral fusiform, calcarine and lingual cortex captured within the MRS voxel?

6. Lines 132-133. Further information should be provided about the factor loadings used for computing the visual sensory factor. How exactly was performance used to determine these loadings?

7. Was participant's visual acuity assessed? If so, please include whether acuity was matched between age groups.

8. Please indicate left and right in the brain images in Figures 2 and 3.

9. Specifying "visual processing" in the title of the paper would be appropriate given the focus on the visual system.

*Reviewer #3 (Recommendations for the authors):*

1. Given the computation of neural variability is the key in this paper, I strongly recommend the authors to detail the calculation of SDBOLD. It is inconvenient to search the reference.

2. Given such a rich dataset and diverse types of analyses, readers easily lost track of the results. I particularly like the first paragraph of the Discussion. It clearly summarizes all the key findings in this paper. I would suggest doing it in the last paragraph of the introduction such that readers can set up expectations and better interpret the follow-up result part.

[Editors' note: further revisions were suggested prior to acceptance, as described below.]

Thank you for resubmitting your work entitled "Modulation of brain signal variability in visual cortex reflects aging, GABA, and behavior" for further consideration by *eLife*. Your revised article has been evaluated by Joshua Gold (Senior Editor) and a Reviewing Editor.

The manuscript has been improved but there are some remaining issues that need to be addressed, as outlined below:

1) Further experimental details on the visual display, the MRS water reference, plus example MRS spectra need to be added to the manuscript.

2) Demonstration of regional specificity of the MRS results to the ventral visual cortex voxel has been requested.

*Reviewer #1 (Recommendations for the authors):*

The authors have done a thorough job of addressing the comments. My comments are below.

Line 548: "where the laptop was placed at a standard" needs to state the exact viewing distance of the screen to the participant.

Example spectra, at group level if possible, should be added to the Supplementary materials to indicate the quality of the MRS.

How specific are the GABA+ effects to the visual cortex voxels? It appears that they should be highly specific, given the visual task, but the evidence has not been supplied. Given the number of additional voxels that have been collected, it seems straightforward to assess region specificity. Could the authors please use a control region from one of the other MRS voxels to provide support for a region specificity of the GABA+ results in the ventral visual cortex, for example, the right sensorimotor cortex?

*Reviewer #2 (Recommendations for the authors):*

The authors have done a commendable job addressing the majority of my concerns.

Regarding my previous comment #2, I am still unclear on how water was measured. I found where the water peak fitting was described (Ln169-170). Presumably, this was using the residual water signal in the edited spectra since a separate unsuppressed water scan was not acquired. It's possible this was explained more completely in the supplements, but I am having trouble finding the supplemental materials that accompany the revised manuscript. Could you please clarify whether the residual water signal is what was used? This is important given that the raw GABA+/H2O data were reported. Thanks.

---

## [Author Response]

Essential revisions (for the authors):Reviewer #1 (Recommendations for the authors):Greater effort can be made to make this paper of broader appeal. In the introduction unpack more the specific approach and relate it to more common measures of the hemodynamic response. The MRS element appears generally well conducted, although basic information on data acquisition and spectral quality is missing.

Thank you for this important point. We have now included an additional discussion about what is known about the more common univariate/mean-BOLD measure, how it relates to BOLD variability, and why modulation of visual variability is of broad interest (Ln 32-40). Briefly, patterns observed in BOLD signal variability are rarely driven by mean-BOLD differences in our entire body of previous work (see Garrett et al., 2013, *NBR*; Waschke et al., 2021, *Neuron*). To further confirm this in the current study, we performed an additional mean-BOLD based analysis and now present these findings (Ln 286-296, Figure 6, Discussion Ln 391-393) to help support the unique contribution that BOLD variability brings to this research context. Results suggest that Δ→ MEAN_BOLD_ was larger in older adults vs. younger adults (Δ→ SD_BOLD_ exhibited the opposite pattern), but more importantly Δ→ MEAN_BOLD_ was not correlated with GABA or with visual performance, while Δ→ SD_BOLD_ was. Such a differentiation is also consistent with prior research (Garrett et.al. 2011, 2013, 2015, 2020) finding SD_BOLD_ to be far more sensitive to behavioral performance than MEAN_BOLD_.

Finally, we have also included a reporting table for Magnetic Resonance Spectroscopy (Table 4) and added more details to the Methods section (Ln 609-623).

– 81. What determined which participants were in which of the 3 subsets? Please name the decision criteria.

The three cohorts were collected at different times as part of an ongoing study whose primary target of interest was the role of GABA in neural dedifferentiation (the Michigan Neural Distinctiveness (MiND) study, [Gagnon et al. (2019). BMC Neurology 19(1), 117]) and that underwent revisions as it progressed. The selection of participants in each cohort was therefore essentially random based on when the participants were recruited in the study timeline (Subset 1: Before Feb 2018, Subset 2: Feb – June 2018, Subset 3: June 2018 – Mar 2020). The revision now clarifies this important point. Thanks.

– What determines the cut-off point for old and young participants?

Young participants were 18 to 29 years old while older participants were 65 and older; they were recruited specifically within these ranges.

– 82. Participants – should include sex. Were participants screened for typical MRS exclusion criteria?

The revision now includes the sex distributions across subsets (Table 3). And yes, participants were screened for several exclusion criteria that are now detailed in the methods and material: various disorders, MRI incompatibility, history of alcohol abuse, etc. (Ln 467-490). Our apologies for not including this information in the original manuscript.

– 89. Does the power analysis refer to a specific subset? It could be subset 3 (38 older and 36 younger) but I am not sure. Please state the sample size and the subset.

Yes, it refers to the behavioral part of the study and we have clarified that in the revision. Ln 492-495: “Since, Garrett et.al. (2020) had found a correlation of r=0.47 between modulation of visual variability and behavioral performance within older adults we had 80% power to detect a correlation of r=0.47 between modulation of variability and performance in each age-group within Subset 3 that completed the behavior task.”

95. I found the text describing the protocol and Figure 1 confusing. All three subsets took part in 3 sessions, but only subset 3 took part in the visual discrimination task. Subset 1 would have taken part in 2 imaging sessions, 1 for the drug and 1 for the placebo. Were those two imaging sessions identical (fMRI + MRS)? This needs to be laid out in detail.– What exactly were the cognitive and behavioural testing components for Subset 1 and 2? Also, please note the average difference in days between sessions 1-3 for cohorts.

We have updated our figure and methods to better clarify the sessions and have included more information about the original study design (Figure 7, Table 3, Ln 504529). Thanks for the suggestion. Other details about the original study that are less relevant to the current paper are described in Gagnon et al. (2019). BMC Neurology 19(1), 1-17.

– Figure 1. I like the use of icons and would suggest an alternative icon to a person with a walking stick to represent people above 77 yo – the alternative could be the same icon as the younger population with a different shading/color.

Great idea, we have done so in Figure 7.

– 104. Not sufficient information to understand what was done here. More information is needed on the testing, such as whether there was a head and chin rest, what was the viewing distance, whether the room was illuminated or dark etc.

Thanks, also done.

Ln 536-539: “Tasks were completed in a well-lit room where the laptop was placed at a standard (same for all participants) comfortable viewing distance. Participants were instructed to not lean forward or squint when completing the tasks.”

– 143. What was the spatial resolution of the EPI scan?

3 × 3 × 3 mm, now included in the revision (Ln 587). Thanks.

– 146. Was the whole fMRI session 6 min- how long did the whole session last and was the fMRI scan always in the same order?

The fMRI sessions lasted approximately 45 min, including the acquisition of a structural image and other task-based and resting state functional data. The order of visual and motor fMRI task-runs was counterbalanced across participants and was always done after the structural, resting-state, and auditory scan. The revision now includes these details too (Ln 599-603). Thanks for the suggestion.

– 154. What was the average hit rate for the target trials, and were there differences across age groups?

There were only 6 target trials that were extremely easy and only designed to ensure participant attention, so almost everyone had a perfect hit rate. We also found no difference between the age-groups for these catch trials (Ln 598).

– 156. MRS session – how were the voxels placed? Did you run another visual task scan to provide a map in the MRS session? The text suggests this is the case, but it is not explicitly mentioned. If you didn't use a localiser map from the same session, please describe how the "person-specific task-activations" were used to guide voxel placement.– What were participants doing during the MRS scan and how many spectral averages were collected? Please include the document "minimum reporting standards" for MRS.

The revision now explains in greater detail how the voxels were placed and other MRS details along with the minimum reporting standards table (Table 4, Ln 609-623). Thanks for the suggestion.

Ln 609-623: “For each participant, we used their fMRI task sessions to compute an activation map for a face and house viewing vs. fixation contrast. These activation maps provided visual guidance to help with the voxel placement. Specifically, the fMRI technician used the activation maps to place the large MRS voxels so that they were centered roughly at the peak of the activation while not extending outside the brain. The voxels were rotated such that one of the edges was almost parallel and close to the cerebellum-cortical boundary minimizing overlap with dura, parietal and angular regions.

It was also rotated in the inferior-posterior plane to maximize coverage in the posterior occipital regions along with the fusiform and parahippocampal gyrus while minimizing overlap with the hippocampus. Note that these are large voxels, and the primary concern is to minimize overlap with non-cortical regions like ventricles, brainstem, dura, and cortex-cerebellum boundary that affect contamination and good shim (FWHM less than 15) while maximizing the coverage of region of interest. In this case the right and left ventrovisual regions were targeted and the voxel covered parts of fusiform, inferior occipital, calcarine, lingual, and parahippocampal gyrus. It was shifted away from the midline as much as possible without covering the dura. The voxel overlap across participants is shown in Figure 1. We used a MEGA-PRESS sequence with the following parameters to obtain MR spectra: TE=68ms (TE1=15ms, TE2=53ms), TR=1.8sec, spec. width=5kHz, Frequency selective editing pulses (14ms) applied at 1.9ppm (ON) & 7.46 ppm (OFF), 256 transients (128 ON interleaved with 128 OFF) of 4096 data points.”

– 168. How was this subset chosen, and how were the drug and placebo randomised? Was the experimenter blind to the randomisation?

The first set of participants to complete the study received the drug-manipulation. The drug and placebo order were counterbalanced across participants. This was only a single-blind study; although the participants did not know whether they were in the drug or placebo condition, the experimenter did (Ln 635-637).

– 173. Why were there two dosages and were low vs high doses balanced?– 183. 7 mm is not the default smoothing in FEAT.

We’ve corrected this error in the revision (Ln 645), thanks.

– 192. Methodological detail missing for quantification of BOLDsd. Please refrain from referring to the previous paper. All information required for replicating the study should be in this current paper.

Apologies, we’ve now added this information to the revision, thanks.

Ln 656-659: “After preprocessing the fMRI data, we normalized all blocks for each condition (houses and faces separately) such that the overall 4D mean across brain and the block was the same. Then, for each voxel, we subtracted the block mean and concatenated the time-series across all blocks for that condition. The standard deviation in this time-series during each of the conditions (faces and houses) was computed at each voxel for each participant (similar to previous work3). Modulation of variability (Δ→ SD_BOLD_) was computed as the difference between the two conditions (SD_BOLD-HOUSES_ – SD_BOLD-FACES_) at each of the voxels in a grey matter volume mask. Similarly, Δ→ SD_BOLD_ on drug was computed for the subset of participants who were administered lorazepam.”

– 208. cite paper for MM contamination in the occipital cortex by Harris et al., 2015.

Done (Ln 442)

– 217. 'quality control check' define and report values.

Also done (Ln 683-684): “Based on a quality control check of the spectra (flagged if the value was more than 3 standard deviations from mean, the fit error was greater than 10% for water, or the CSF fraction was more than 20%), we flagged and discarded three right and seven left GABA values.”

– 263. It isn't clear to me how the measure of BOLDsd relates to conventional glm-approaches that isolate regions with stimulus preference. As far as the data shows, the BOLDsd approach does not isolate object-selective regions. Please lay out in more detail the rationale for using BOLDsd instead of conventional GLM contrast in the introduction; report the result of the glm contrast faces>houses in percentage BOLD-signal change and discuss their results in light of the different analyses approaches. This would also help broaden the accessibility of your article.

The revised Introduction tries to do a better job of motivating the use of BOLDsd and its relationship to more traditional measures of mean signal change (Ln 32-40). We’ve also added the results based on mean BOLD (Ln 286-296, Figure 6, Discussion Ln 391393). Briefly, our results suggest that Δ→ MEAN_BOLD_ is actually larger in older adults vs. younger adults (Δ→ SD_BOLD_ exhibited the opposite pattern), but more importantly, Δ→ MEAN_BOLD_ is not correlated with GABA or with visual performance. This is also consistent with prior research that found MEAN_BOLD_ to be relatively insensitive to behavioral performance (Garrett et.al. 2011, 2013, 2015, 2020).

– 288-289 This sentence struck me as odd and makes the analysis sound post hoc.

Good point. We’ve modified this sentence to try to make it read better.

Now Ln 155-158: “We further confirmed that the strongest task-condition effects on SD_BOLD_ are present in the visual cortex and our GABA estimates are also in the taskrelevant visual cortex. Thus, we restricted all further analyses within an anatomically defined task-relevant visual mask in all subjects (see Methods) rather than investigate it in the whole-brain and in task irrelevant regions.”

– 314: I'm not sure how the data in Figure 4 was pulled together, this comes back to Figure 1 where more detail is needed on what exactly different people were tested on, and in what order. Is the baseline GABA from one of the on/off-drug sessions, or is it separate? I am guessing that the y-axis is the difference between placebo and drug in fMRI, but that the x-axis is from a separate off-drug MRS session. Whatever the case needs to be clarified.

The GABA MRS sessions were separate from the fMRI sessions and were always offdrug. And yes, the x-axis is the GABA estimate from the off-drug MRS session while the y-axis is the difference in Δ→ SD_BOLD_ on and off drug. We tried to clarify this in the revision. Thanks for the suggestion (Ln 186-188 and Figure 4).

– GABA levels should be noted with units on the axis GABA+/water IU.

Done, thanks (Figure 3a, 4 and 5a).

– 319: Why is the change due to drug consistent with an inverted-U account if the change simply describes a negative relationship of BOLDsd to baseline GABA?

The drug-related shift in Δ→ SD_BOLD_ extends from +15 to -15 and has a negative relationship with baseline GABA. Thus, participants with low baseline GABA had low levels of Δ→ SD_BOLD_ and experienced an increase in Δ→ SD_BOLD_ on the drug (i.e., when GABA activity was increased). However, participants with higher baseline GABA levels and higher Δ→ SD_BOLD_ experienced a no-change or even a decrease in Δ→ SD_BOLD_ on drug (i.e. when GABA was increased beyond optimal levels). This is consistent with the inverted-U account. We have added an additional explanation on Ln 355-362 in manuscript as well.

**Author response image 1. sa2fig1:** 

Reviewer #2 (Recommendations for the authors):1. Baseline GABA+ and the task-condition effect on BOLD variability differed between age groups. Combining the two groups to then test for a relationship between GABA and the task-condition effect on BOLD variability can suggest a relationship that may be explained by age alone. The analyses should be repeated with age as a nuisance regressor. Along these lines, Figure 3 shows separate trend lines for the two groups, but the effect was reported across both groups in the text. The figure implies that both groups independently exhibit a relationship, but that is not the case.

We have now clarified in the text that indeed the relationship between GABA and Δ→ SD_BOLD_ shown in Figure 3 is significant within each age-group (Line 173-175). Further, the initial, full model included both age-group and GABA as regressors and found that both had a significant effect (while the Age-group x GABA interaction was not significant). Therefore, the effect of age on Δ→ SD_BOLD_ does not completely explain the observed relationship between GABA and Δ→ SD_BOLD,_ as this effect is significant in both age-groups individually and in the whole sample even when variance explained by age is accounted for. The revision tries to make these points clearer (Ln 331-335).

2. Further information should be included about the MRS imaging approach. (1) What were the shim values for the voxels in the different groups? (2) What was the quality control (line 217)? (3) How were voxels positioned beyond centering over peak activation? Figure 1 suggests voxels were rotated in at least two planes based on anatomy. This information is missing from the manuscript. (4) A separate water peak estimate was conducted as described in the supplements, but it is not clear whether an additional PRESS scan was acquired. Please clarify.

The revision now includes all details to address these concerns as well as the table of "minimum reporting standards" for MRS has been included in the supplementary attachments (Table 4). No additional water-unsuppressed PRESS scan was obtained.

Ln 609 – 623 – The MRS data was obtained from 3cm x 3cm x 3cm voxels placed in the left and right ventrovisual cortex. For each participant, we used their fMRI task sessions to compute an activation map for a face and house viewing vs. fixation contrast. These activation maps provided visual guidance to help with the voxel placement. Specifically, the MRI technician used the activation maps to place the large MRS voxels so that they were centered roughly at the peak of the activation while not extending outside the brain. The voxels were rotated such that one of the edges was almost parallel and close to the cerebellum-cortical boundary minimizing overlap with dura, parietal and angular regions. It was also rotated in the inferior-posterior plane to maximize coverage in the posterior occipital regions along with the fusiform and parahippocampal gyrus while minimizing overlap with the hippocampus. Note that these are large voxels, and the primary concern is to minimize overlap with non-cortical regions like ventricles, brainstem, dura, and cortex-cerebellum boundary that affect contamination and good shim (FWHM less than 15) while maximizing the coverage of region of interest. In this case the right and left ventrovisual regions were targeted and the voxel covered parts of fusiform, inferior occipital, calcarine, lingual, and parahippocampal gyrus. It was shifted away from the midline as much as possible without covering the dura. The voxel overlap across participants is shown in Figure 7.

Ln 683-685 – Based on a quality control check of the spectra (flagged if the value was more than 3 standard deviations from the mean, the fit error was greater than 10% for water, or the CSF fraction was more than 20%), we flagged and discarded three right and seven left GABA values.

5) Move the supplemental information about tissue-corrected GABA measurements to the main text. Otherwise, lines 212 to 221 make it somewhat ambiguous as to which GABA measurement was used in subsequent analyses.

Done, thanks.

Ln 148-155: “GABA+ levels were also significantly lower in older adults compared to younger adults after correcting for tissue-composition differences (t(131.9) = -3.13, p = 0.002, Cohen’s d = 0.53) after applying Harris et al.'s α tissue correction procedure. These results suggest that the differences in tissue composition do not explain the observed lower GABA levels in older adults. Since, raw GABA+/H2O levels were highly correlated with α tissue composition-corrected GABA levels (r(131) = 0.88, p < 2.2e-16), all results presented below were based on the raw GABA+/H2O levels.”

3. Lines 173-174. The lorazepam manipulation is an important feature of the study. However, it is not clear why the two different dosages (.5 or 1 mg) were used and how many participants from each age group received which dosages. Also, were these dosages administered equally across individuals with different baseline GABA levels? Additional analyses should be reported to address whether the different dosages had different effects on BOLD variability across age groups and baseline GABA levels.

The dosage was assigned randomly (Ln 514, 635) and we did not find any effect of dosage (Ln 195-211).

4. Lines 158-160. Please include the scanning parameters for the T1-weighted MRI in the Methods section even if they are reported in a previous publication.

Done.

Ln 578-582: “A high-resolution T1-weighted structural image was collected using a 3D fast spoiled gradient echo (SPGR) BRAVO pulse sequence with the following parameters: TR = 12.2 ms; TE = 5.2 ms; TI = 500 ms; flip angle = 15°; FOV = 256 × 256 mm; 156 axial slices with thickness = 1 mm and no spacing; acquisition time = 5 min and voxel size (1 × 1 × 1 mm).”

5. Lines 308-310. Were the bilateral fusiform, calcarine and lingual cortex captured within the MRS voxel?

Yes (Now included Ln 609-623). For each participant, we used their fMRI task sessions to compute an activation map for a face and house viewing vs. fixation contrast. These activation maps provided a visual guidance to help with the voxel placement where the fMRI technician used major anatomical landmarks to place the large MRS voxels centered roughly at the peak of the activation. The voxels were rotated such that one of the edges was almost parallel and close to the cerebellum-cortical boundary minimizing overlap with dura, parietal and angular regions. It was also rotated in the inferiorposterior plane to maximize coverage in the posterior occipital regions along with fusiform and the parahippocampus while minimizing overlap with hippocampus. Note that these are large voxels, and the primary concern is to minimize overlap with noncortical regions like ventricles, brainstem, dura, and cortex-cerebellum boundary that affect contamination and good shim (FWHM less than 15) while maximizing the coverage of region of interest. In this case the right and left ventrovisual regions were targeted and the voxel covered parts of fusiform, infereior occipital region, calcarine, lingual, and parahippocampus. It was shifted away from the midline as much as possible without covering the dura. The voxel overlap across participants is shown in Figure 7.

6. Lines 132-133. Further information should be provided about the factor loadings used for computing the visual sensory factor. How exactly was performance used to determine these loadings?

Done. Each task used a staircase procedure in which more noise was presented after correct trials and less noise was presented after incorrect trials in order to estimate the amount of noise the participant could tolerate. The average level of noise presented for the last 40 trials for each of four task was used as the dependent measure.

Now Ln 566-582: “Using the MATLAB function factoran, we computed a single visual sensory factor based on this measure from each of the four tasks. The loadings of the tasks on the factor were 0.3 (buildings), 0.8 (faces), 0.5 (objects) and 0.2 (scenes). Since, a higher score on each individual task represents better performance and each loading is positive a higher factor score also reflects better performance. Using principal component analysis (PCA) only one component had an eigen value > 1, explaining 40% of the variance, and the corresponding weights were 0.53 (buildings), 0.8 (faces), 0.7 (objects) and 0.42 (scenes). The scores on this first component were correlated at (r(72) = 0.88, p < 2.2e-16) with the scores on the first factor from the factor analysis. We thus present results based on the factor score.”

7. Was participant's visual acuity assessed? If so, please include whether acuity was matched between age groups.

Good point. We have now included additional analysis (Ln 273-283). Briefly, visual acuity was assessed using the NIH-toolbox and was significantly lower in older adults compared to young. However, controlling for it did not affect the reported findings. Raw visual acuity was significantly lower in older adults (t(56.5) = 3.85, p = 0.0002) as measured by the NIH Visual Acuity task, but it was not correlated with the visual discrimination behavior score (r(72) = -0.03, p = 0.78). This is not surprising as the discrimination task was performed with images presented at a comfortable distance and size to the participant. Moreover, controlling for visual acuity did not alter any of the reported effects. The relationship between both age (F(1,119) = 67.01, p < 3.4e-13) and GABA (F(1,119) = 5.9, p = 0.016) remained robustly associated with visual Δ→ SD_BOLD_ after controlling for visual acuity and grey matter volume. Moreover, both greater Δ→ SD_BOLD_ (rho (36) = 0.41, p = 0.01) and higher GABA levels (rho (36) = 0.35, p = 0.03) were associated with better visual discrimination performance in older adults even after controlling for visual acuity and grey matter volume.

8. Please indicate left and right in the brain images in Figures 2 and 3.

Done, thanks for the suggestion.

9. Specifying "visual processing" in the title of the paper would be appropriate given the focus on the visual system.

Also done.

Reviewer #3 (Recommendations for the authors):1. Given the computation of neural variability is the key in this paper, I strongly recommend the authors to detail the calculation of SDBOLD. It is inconvenient to search the reference.

Done (Ln 656-663), thanks for the suggestion. After preprocessing the fMRI data, we normalized all blocks for each condition (houses and faces separately) such that the overall 4D mean across brain and the block was the same. Then, for each voxel, we subtracted the block mean and concatenated the time-series across all blocks for that condition. The standard deviation in this time-series during each of the conditions (faces and houses) was computed at each voxel for each participant. Modulation of variability (Δ→ SD_BOLD_) was computed as the difference between the two conditions (SD_BOLD-Houses_ – SD_BOLD-Faces_) at each of the voxels in a grey matter volume mask.

2. Given such a rich dataset and diverse types of analyses, readers easily lost track of the results. I particularly like the first paragraph of the Discussion. It clearly summarizes all the key findings in this paper. I would suggest doing it in the last paragraph of the introduction such that readers can set up expectations and better interpret the follow-up result part.

Done (Ln 79-84), thanks.

[Editors’ note: what follows is the authors’ response to the second round of review.]

The manuscript has been improved but there are some remaining issues that need to be addressed, as outlined below:Reviewer #1 (Recommendations for the authors):The authors have done a thorough job of addressing the comments. My comments are below.Line 548: "where the laptop was placed at a standard" needs to state the exact viewing distance of the screen to the participant.

Now updated on Line 551 “laptop was placed at a fixed (same and measured for all participants) viewing distance of 21 inches.”

Example spectra, at group level if possible, should be added to the Supplementary materials to indicate the quality of the MRS.How specific are the GABA+ effects to the visual cortex voxels? It appears that they should be highly specific, given the visual task, but the evidence has not been supplied. Given the number of additional voxels that have been collected, it seems straightforward to assess region specificity. Could the authors please use a control region from one of the other MRS voxels to provide support for a region specificity of the GABA+ results in the ventral visual cortex, for example, the right sensorimotor cortex?

Great point. Indeed, the relationship between ventral visual cortex GABA+ levels and modulation of variability is region specific. Modulation of variability in the visual cortex during a visual task was not associated with the GABA levels in the auditory cortex or motor cortex. On lines 182-189, we now note that:

Further, we found that tissue-composition corrected auditory GABA+ levels (see Lalwani et.al., 2019 for details of voxel placement) were not associated with modulation of visual variability in the whole sample after controlling for age (F(1,120) = 0.72, p = 0.4) or within older adults alone (r(74) = 0.01, p = 0.9). Similarly, tissue-composition corrected motor GABA+ levels (see Cassady et.al., 2019 for details of voxel placement) were not associated with modulation of visual variability in the whole sample after controlling for age (F(1,120) = 0.52, p = 0.47) or within older adults alone (r(74) = 0.02, p = 0.9), revealing that our GABA+ effects were specific to visual cortex.

On lines 344-346, we now note:

Moreover, we found that this relationship was specific to visual cortex (i.e., auditory and motor GABA levels did not predict individual differences in visual ∆SD_BOLD)_.

Reviewer #2 (Recommendations for the authors):The authors have done a commendable job addressing the majority of my concerns.Regarding my previous comment #2, I am still unclear on how water was measured. I found where the water peak fitting was described (Ln169-170). Presumably, this was using the residual water signal in the edited spectra since a separate unsuppressed water scan was not acquired. It's possible this was explained more completely in the supplements, but I am having trouble finding the supplemental materials that accompany the revised manuscript. Could you please clarify whether the residual water signal is what was used? This is important given that the raw GABA+/H2O data were reported. Thanks.

Thank you so much for pointing out this. The previous revision was unclear, and we have now updated it appropriately. “Line 641: For water referencing, eight transients of water unsuppressed data were collected at the end of the sequence, resulting in one reference frame where the NEX=8 averaging is applied.”